# RANDOM MESH PROJECTORS FOR INVERSE PROBLEMS

**Konik Kothari**[*]
University of Illinois at Urbana-Champaign
kkothar3@illinois.edu

**Sidharth Gupta**[*]
University of Illinois at Urbana-Champaign
gupta67@illinois.edu

**Maarten V. de Hoop**
Rice University
mdehoop@rice.edu

**Ivan Dokmanić**
University of Illinois at Urbana-Champaign
dokmanic@illinois.edu

## ABSTRACT

We propose a new learning-based approach to solve ill-posed inverse problems in imaging. We address the case where ground truth training samples are rare and the problem is severely ill-posed—both because of the underlying physics and because we can only get few measurements. This setting is common in geophysical imaging and remote sensing. We show that in this case the common approach to directly learn the mapping from the measured data to the reconstruction becomes unstable. Instead, we propose to first learn an ensemble of simpler mappings from the data to projections of the unknown image into random piecewise-constant subspaces. We then combine the projections to form a final reconstruction by solving a deconvolution-like problem. We show experimentally that the proposed method is more robust to measurement noise and corruptions not seen during training than a directly learned inverse.

## 1 INTRODUCTION

A variety of imaging inverse problems can be discretized to a linear system $\boldsymbol{y} = \boldsymbol{Ax} + \boldsymbol{\eta}$ where $\boldsymbol{y} \in \mathbb{R}^M$ is the measured data, $\boldsymbol{A} \in \mathbb{R}^{M \times N}$ is the imaging or forward operator, $\boldsymbol{x} \in \mathcal{X} \subset \mathbb{R}^N$ is the object being probed by applying $\boldsymbol{A}$ (often called the model), and $\boldsymbol{\eta}$ is the noise. Depending on the application, the set of plausible reconstructions $\mathcal{X}$ could model natural, seismic, or biomedical images. In many cases the resulting inverse problem is ill-posed, either because of the poor conditioning of $\boldsymbol{A}$ (a consequence of the underlying physics) or because $M \ll N$.

A classical approach to solve ill-posed inverse problems is to minimize an objective functional regularized via a certain norm (e.g. $\ell_1$, $\ell_2$, total variation (TV) seminorm) of the model. These methods promote general properties such as sparsity or smoothness of reconstructions, sometimes in combination with learned synthesis or analysis operators, or dictionaries (Sprechmann et al. (2013)).

In this paper, we address situations with very sparse measurement data ($M \ll N$) so that even a coarse reconstruction of the unknown model is hard to get with traditional regularization schemes. Unlike artifact-removal scenarios where applying a regularized pseudoinverse of the imaging operator already brings out considerable structure, we look at applications where standard techniques cannot produce a reasonable image (Figure 1). This highly unresolved regime is common in geophysics and requires alternative, more involved strategies (Galetti et al. (2017)).

An appealing alternative to classical regularizers is to use deep neural networks. For example, generative models (GANs) based on neural networks have recently achieved impressive results in regularization of inverse problems (Bora et al. (2018), Lunz et al. (2018)). However, a difficulty in geophysical applications is that there are very few examples of ground truth models available for training (sometimes none at all). Since GANs require many, they cannot be applied to such problems. This suggests to look for methods that are not very sensitive to the training dataset. Conversely, it means that the sought reconstructions are less detailed than what is expected in data-rich settings; for

---

[*]S. Gupta and K. Kothari contributed equally.

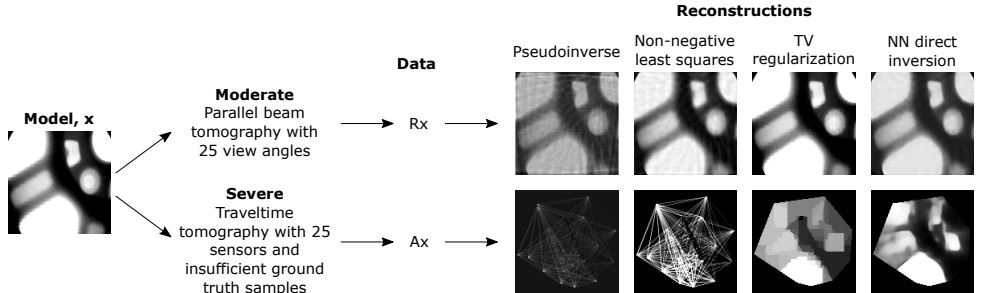

Figure 1: We reconstruct an image $x$ from its tomographic measurements. In moderately ill-posed problems, conventional methods based on the pseudoinverse and regularized non-negative least squares ($x \in [0, 1]^N$, $N$ is image dimension) give correct structural information. In fact, total variation (TV) approaches give very good results. A neural network (Jin et al. (2016)) can be trained to directly invert and remove the artifacts (NN). In a severely ill-posed problem on the other hand (explained in Figure 4) with insufficient ground truth training data, neither the classical techniques nor a neural network recover salient geometric features.

an example, see the reconstructions of the Tibetan plateau (Yao et al. (2006)).

In this paper, we propose a two-stage method to solve ill-posed inverse problems using random low-dimensional projections and convolutional neural networks. We first decompose the inverse problem into a collection of simpler learning problems of estimating projections into random (but structured) low-dimensional subspaces of piecewise-constant images. Each projection is easier to learn in terms of generalization error (Cooper (1995)) thanks to its lower Lipschitz constant.

In the second stage, we solve a new linear inverse problem that combines the estimates from the different subspaces. We show that this converts the original problem with possibly non-local (often tomographic) measurements into an inverse problem with localized measurements, and that in fact, in expectation over random subspaces the problem becomes a deconvolution. Intuitively, projecting into piecewise-constant subspaces is equivalent to estimating local averages—a simpler problem than estimating individual pixel values. Combining the local estimates lets us recover the underlying structure. We believe that this technique is of independent interest in addressing inverse problems.

We test our method on linearized seismic traveltime tomography (Bording et al. (1987); Hole (1992)) with sparse measurements and show that it outperforms learned direct inversion in quality of achieved reconstructions, robustness to measurement errors, and (in)sensitivity to the training data. The latter is essential in domains with insufficient ground truth images.

## 2  RELATED WORK

Although neural networks have long been used to address inverse problems (Ogawa et al. (1998); Hoole (1993); Schiller and Doerffer (2010)), the past few years have seen the number of related deep learning papers grow exponentially. The majority address biomedical imaging (Güler and Übeyli (2005); Hudson and Cohen (2000)) with several special issues[1] and review papers (Lucas et al. (2018); McCann et al. (2017)) dedicated to the topic. All these papers address reconstruction from subsampled or low-quality data, often motivated by reduced scanning time or lower radiation doses. Beyond biomedical imaging, machine learning techniques are emerging in geophysical imaging (Araya-Polo et al. (2017); Lewis and Vigh (2017); Bianco and Gertoft (2017)), though at a slower pace, perhaps partly due to the lack of standard open datasets.

Existing methods can be grouped into non-iterative methods that learn a feed-forward mapping from the measured data $y$ (or some standard manipulation such as adjoint or a pseudoinverse) to the model $x$ (Jin et al. (2016); Pelt and Batenburg (2013); Zhu et al. (2018); Wang (2016); Antholzer et al. (2017); Han et al. (2016); Zhang et al. (2016)); and iterative energy minimization methods, with

---

[1]IEEE Transactions on Medical Imaging, May 2016 (Greenspan et al. (2016)); IEEE Signal Processing Magazine, November 2017, January 2018 (Porikli et al. (2017; 2018)).

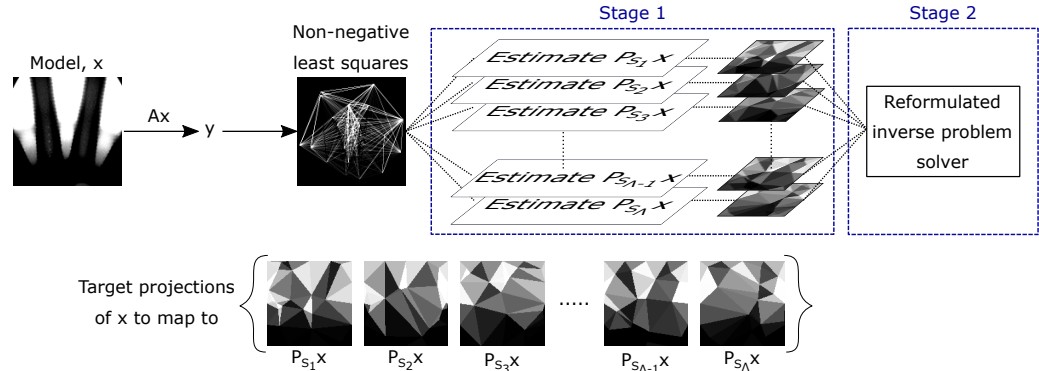

Figure 2: Regularization by $\Lambda$ random projections: 1) each orthogonal projection is approximated by a convolutional neural network which maps from a non-negative least squares reconstruction of an image to its projection onto a lower dimension subspace of Delaunay triangulations; 2) projections are combined to estimate the original image using regularized least squares.

either the regularizer being a neural network (Li et al. (2018)), or neural networks replacing various iteration components such as gradients, projectors, or proximal mappings (Kelly et al. (2017); Adler and Öktem (2017b;a); Chang et al. (2017)). These are further related to the notion of plug-and-play regularization (Venkatakrishnan et al. (2013)), as well as early uses of neural nets to unroll and adapt standard sparse reconstruction algorithms (Gregor and LeCun (2010); Xin et al. (2016)). An advantage of the first group of methods is that they are fast; an advantage of the second group is that they are better at enforcing data consistency.

**Generative models**   A rather different take was proposed in the context of compressed sensing where the reconstruction is constrained to lie in the range of a pretrained generative network (Bora et al. (2017; 2018)). Their scheme achieves impressive results on random sensing operators and comes with theoretical guarantees. However, training generative networks requires many examples of ground truth and the method is inherently subject to dataset bias. Here, we focus on a setting where ground-truth samples are very few or impossible to obtain.

There are connections between our work and sketching (Gribonval et al. (2017); Pilanci and Wainwright (2016)) where the learning problem is also simplified by random low-dimensional projections of some object—either the data or the unknown reconstruction itself (Yurtsever et al. (2017)). This also exposes natural connections with learning via random features (Rahimi and Recht (2008; 2009)).

## 3 REGULARIZATION BY RANDOM MESH PROJECTIONS

The two stages of our method are **(i)** decomposing a "hard" learning task of directly learning an unstable operator into an ensemble of "easy" tasks of estimating projections of the unknown model into low-dimensional subspaces; and **(ii)** combining these projection estimates to solve a reformulated inverse problem for $\boldsymbol{x}$. The two stages are summarized in Figure 2. While our method is applicable to continuous and non-linear settings, we focus on linear finite-dimensional inverse problems.

### 3.1 DECOMPOSING THE LEARNING PROBLEM

Statistical learning theory tells us that the number of samples required to learn an $M$-variate $L$-Lipschitz function to a given sup-norm accuracy is $\mathcal{O}(L^M)$ (Cooper (1995)). While this result is proved for scalar-valued multivariate maps, it is reasonable to expect the same scaling in $L$ to hold for vector-valued maps. This motivates us to study Lipschitz properties of the projected inverse maps.

We wish to reconstruct $\boldsymbol{x}$, an $N$-pixel image from $\mathcal{X} \subset \mathbb{R}^N$ where $N$ is large (we think of $\boldsymbol{x}$ as an $\sqrt{N} \times \sqrt{N}$ discrete image). We assume that the map from $\boldsymbol{x} \in \mathcal{X}$ to $\boldsymbol{y} \in \mathbb{R}^M$ is injective so that it is invertible on its range, and that there exists an $L$-Lipschitz (generally non-linear) inverse $G$,

$$\|G(\boldsymbol{y}_1) - G(\boldsymbol{y}_2)\| \leq L \|\boldsymbol{y}_1 - \boldsymbol{y}_2\|.$$

In order for the injectivity assumption to be reasonable, we assume that $\mathcal{X}$ is a low-dimensional manifold embedded in $\mathbb{R}^N$ of dimension at most $M$, where $M$ is the number of measurements. Since we are in finite dimension, injectivity implies the existence of $L$ (Stefanov and Uhlmann (2009)). Due to ill-posedness, $L$ is typically large.

Consider now the map from the data $\boldsymbol{y}$ to a projection of the model $\boldsymbol{x}$ into some $K$-dimensional subspace $S$, where $K \ll N$. Note that this map exists by construction (since $A$ is injective on $\mathcal{X}$), and that it must be non-linear. To see this, note that the only consistent[2] linear map acting on $\boldsymbol{y}$ is an oblique, rather than an orthogonal projection on $S$ (cf. Section 2.4 in Vetterli et al. (2014)). We explain this in more detail in Appendix A.

Denote the projection by $\boldsymbol{P}_S \boldsymbol{x}$ and assume $S \subset \mathbb{R}^N$ is chosen uniformly at random.[3] We want to evaluate the expected Lipschitz constant of the map from $\boldsymbol{y}$ to $\boldsymbol{P}_S \boldsymbol{x}$, noting that it can be written as $\boldsymbol{P}_S \bullet G$:

$$\mathbb{E} \ \|\boldsymbol{P}_S \bullet G(\boldsymbol{y}_1) - \boldsymbol{P}_S \bullet G(\boldsymbol{y}_2)\| \le \sqrt{\mathbb{E} \ \|\boldsymbol{P}_S \bullet G(\boldsymbol{y}_1) - \boldsymbol{P}_S \bullet G(\boldsymbol{y}_2)\|^2} \le \sqrt{\tfrac{K}{N}} L \ \|\boldsymbol{y}_1 - \boldsymbol{y}_2\|$$

where the first inequality is Jensen's inequality, and the second one follows from

$$\mathbb{E} \|\boldsymbol{P}_S \boldsymbol{x}\|^2 = \mathbb{E} \, \boldsymbol{x}^\top \boldsymbol{P}_S^\top \boldsymbol{P}_S \boldsymbol{x} = \boldsymbol{x}^\top \mathbb{E}(\boldsymbol{P}_S^\top \boldsymbol{P}_S) \boldsymbol{x}$$

and the observation that $\mathbb{E} \, \boldsymbol{P}_S^\top \boldsymbol{P}_S = \frac{K}{N} \boldsymbol{I}_N$. In other words, random projections reduce the Lipschitz constant by a factor of $\sqrt{K/N}$ on average. Since learning requires $\mathcal{O}(L^K)$ samples, this allows us to work with exponentially fewer samples and makes the learning task easier. Conversely, given a fixed training dataset, it gives more accurate estimates.

### 3.1.1 A case for Delaunay triangulations

The above example uses unstructured random subspaces. In many inverse problems, such as inverse scattering (Beretta et al. (2013); Di Cristo and Rondi (2003)), a judicious choice of subspace family can give exponential improvements in Lipschitz stability. Particularly, it is favorable to use piecewise-constant images: $\boldsymbol{x} = \sum_{k=1}^{K} \boldsymbol{x}_k \boldsymbol{\chi}_k$, with $\boldsymbol{\chi}_k$ being indicator functions of some domain subset.

Motivated by this observation, we use piecewise-constant subspaces over random Delaunay triangle meshes. The Delaunay triangulations enjoy a number of desirable learning-theoretic properties. For function learning it was shown that given a set of vertices, piecewise linear functions on Delaunay triangulations achieve the smallest sup-norm error among all triangulations (Omohundro (1989)).

We sample $\Lambda$ sets of points in the image domain from a uniform-density Poisson process and construct $\Lambda$ (discrete) Delaunay triangulations with those points as vertices. Let $\mathcal{S} = \{S_\lambda \mid 1 \le \lambda \le \Lambda\}$ be the collection of $\Lambda$ subspaces of piecewise-constant functions on these triangulations. Let further $G_\lambda$ be the map from $\boldsymbol{y}$ to the projection of the model into subspace $S_\lambda$, $G_\lambda \boldsymbol{y} = \boldsymbol{P}_{S_\lambda} \boldsymbol{x}$. Instead of learning the "hard" inverse mapping $G$, we propose to learn an ensemble of simpler mappings $\{G_\lambda\}_{\lambda=1}^{\Lambda}$.

We approximate each $G_\lambda$ by a convolutional neural network, $\Gamma_{\theta(\lambda)}(\widetilde{\boldsymbol{y}}) : \mathbb{R}^N \to \mathbb{R}^N$, parameterized by a set of trained weights $\theta(\lambda)$. Similar to Jin et al. (2016), we do not use the measured data $\boldsymbol{y} \in \mathbb{R}^M$ directly as this would require the network to first learn to map $\boldsymbol{y}$ back to the image domain; we rather warm-start the reconstruction by a non-negative least squares reconstruction, $\widetilde{\boldsymbol{y}} \in \mathbb{R}^N$, computed from $\boldsymbol{y}$. The weights are chosen by minimizing empirical risk:

$$\theta(\lambda) = \arg\min_{\theta} \frac{1}{J} \sum_{j=1}^{J} \left\|\Gamma_{\theta(\lambda)}(\widetilde{\boldsymbol{y}}_j) - \boldsymbol{P}_{S_\lambda} \boldsymbol{x}_j\right\|_2^2, \tag{1}$$

where $\left\{(\boldsymbol{x}_j, \widetilde{\boldsymbol{y}}_j)\right\}_{j=1}^{J}$ is a set of $J$ training models and non-negative least squares measurements.

### 3.2 The new inverse problem

By learning projections onto random subspaces, we transform our original problem into that of estimating $\boldsymbol{x}$ from $\left\{\Gamma_{\theta(\lambda)}(\widetilde{\boldsymbol{y}})\right\}_{\lambda=1}^{\Lambda}$. To see how this can be done, ascribe to the columns of $\boldsymbol{B}_\lambda \in$

---

[2]Consistent meaning that if $\boldsymbol{x}$ already lives in $S$, then the map should return $\boldsymbol{x}$.

[3]One way to construct the corresponding projection matrix is as $\boldsymbol{P}_S = \boldsymbol{W} \boldsymbol{W}^\dagger$, where $\boldsymbol{W} \in \mathbb{R}^{N \times K}$ is a matrix with standard iid Gaussian entries.

$\mathbb{R}^{N \times K}$ a natural orthogonal basis for the subspace $S_\lambda$, $\boldsymbol{B}_\lambda = [\boldsymbol{\chi}_{\lambda,1}, \ldots, \boldsymbol{\chi}_{\lambda,K}]$, with $\boldsymbol{\chi}_{\lambda,k}$ being the indicator function of the $k$th triangle in mesh $\lambda$. Denote by $\boldsymbol{q}_\lambda \overset{\text{def}}{=} \boldsymbol{q}_\lambda(\boldsymbol{y})$ the mapping from the data $\boldsymbol{y}$ to an estimate of the expansion coefficients of $\boldsymbol{x}$ in the basis for $S_\lambda$:

$$\boldsymbol{q}_\lambda(\boldsymbol{y}) \overset{\text{def}}{=} \boldsymbol{B}_\lambda^\top \Gamma_{\theta(\lambda)}(\widetilde{\boldsymbol{y}})$$

Let $\boldsymbol{B} \overset{\text{def}}{=} \begin{bmatrix} \boldsymbol{B}_1 \ \boldsymbol{B}_2 \ \ldots \ \boldsymbol{B}_\Lambda \end{bmatrix} \in \mathbb{R}^{N \times K\Lambda}$, and $\boldsymbol{q} \overset{\text{def}}{=} \boldsymbol{q}(y) \overset{\text{def}}{=} \begin{bmatrix} \boldsymbol{q}_1^\top, \boldsymbol{q}_2^\top, \ldots, \boldsymbol{q}_\Lambda^\top \end{bmatrix}^\top \in \mathbb{R}^{K\Lambda}$; then we can estimate $\boldsymbol{x}$ using the following reformulated problem:

$$\boldsymbol{q} \approx \boldsymbol{B}^\top \boldsymbol{x},$$

and the corresponding regularized reconstruction:

$$\widehat{\boldsymbol{x}} = \widehat{G}(\boldsymbol{y}) \overset{\text{def}}{=} \underset{\boldsymbol{x} \in [0,1]^N}{\arg\min} \ \left\| \boldsymbol{q}(\boldsymbol{y}) - \boldsymbol{B}^\top \boldsymbol{x} \right\|^2 + \lambda \varphi(\boldsymbol{x}), \tag{2}$$

with $\varphi(\boldsymbol{x})$ chosen as the TV-seminorm $\|\boldsymbol{x}\|_{\text{TV}}$. The regularization is not essential. As we show experimentally, if $K\Lambda$ is sufficiently large, $\varphi(\boldsymbol{x})$ is not required. Note that solving the original problem directly using $\|\boldsymbol{x}\|_{\text{TV}}$ regularizer fails to recover the structure of the model (Figure 1).

### 3.3 STABILITY OF THE REFORMULATED PROBLEM AND "CONVOLUTIONALIZATION"

Since the true inverse map $G$ has a large Lipschitz constant, it would seem reasonable that as the number of mesh subspaces $\Lambda$ grows large (and their direct sum approaches the whole ambient space $\mathbb{R}^N$), the Lipschitz properties of $\widehat{G}$ should deteriorate as well.

Denote the unregularized inverse mapping in $\boldsymbol{y} \mapsto \widehat{\boldsymbol{x}}$ (2) by $\overline{G}$. Then we have the following estimate:

$$\left\| \overline{G}(\boldsymbol{y}_1) - \overline{G}(\boldsymbol{y}_2) \right\| = \left\| (\boldsymbol{B}^T)^\dagger \boldsymbol{q}(\boldsymbol{y}_1) - (\boldsymbol{B}^T)^\dagger \boldsymbol{q}(\boldsymbol{y}_2) \right\| \leq \sigma_{\min}(\boldsymbol{B})^{-1} \sqrt{\Lambda} L_K \left\| \boldsymbol{y}_1 - \boldsymbol{y}_2 \right\|,$$

with $\sigma_{\min}(\boldsymbol{B})$ the smallest (non-zero) singular value of $\boldsymbol{B}$ and $L_K$ the Lipschitz constant of the stable projection mappings $\boldsymbol{q}_\lambda$. Indeed, we observe empirically that $\sigma_{\min}(\boldsymbol{B})^{-1}$ grows large as the number of subspaces increases which reflects the fact that although individual projections are easier to learn, the full-resolution reconstruction remains ill-posed.

Estimates of individual subspace projections give correct *local* information. They convert possibly non-local measurements (e.g. integrals along curves in tomography) into local ones. The key is that these local averages (subspace projection coefficients) can be estimated accurately (see Section 4).

To further illustrate what we mean by correct local information, consider a simple numerical experiment with our reformulated problem, $\boldsymbol{q} = \boldsymbol{B}^T \boldsymbol{x}$, where $\boldsymbol{x}$ is an all-zero image with a few pixels "on". For the sake of clarity we assume the coefficients $\boldsymbol{q}$ are perfect. Recall that $\boldsymbol{B}$ is a block matrix comprising $\Lambda$ subspace bases stacked side by side. It is a random matrix because the subspaces are generated at random, and therefore the reconstruction $\widehat{\boldsymbol{x}} = (\boldsymbol{B}^\top)^\dagger \boldsymbol{q}$ is also random. We approximate $\mathbb{E}\,\widehat{\boldsymbol{x}}$ by simulating a large number of $\Lambda$-tuples of meshes and averaging the obtained reconstructions.

Results are shown in Figure 3 for different numbers of triangles per subspace, $K$, and subspaces per reconstruction, $\Lambda$. As $\Lambda$ or $K$ increase, the expected reconstruction becomes increasingly localized around non-zero pixels. The following proposition (proved in Appendix B) tells us that this phenomenon can be modeled by convolution.[4]

**Proposition 1.** *Let $\widehat{\boldsymbol{x}}$ be the solution to $\boldsymbol{q} = \boldsymbol{B}^\top \boldsymbol{x}$ given as $(\boldsymbol{B}^\top)^\dagger \boldsymbol{q}$. Then there exists a kernel $\widetilde{\boldsymbol{\kappa}}(u)$, with $u$ a discrete index, such that $\mathbb{E}\,\widehat{\boldsymbol{x}} = \boldsymbol{x} * \widetilde{\boldsymbol{\kappa}}$. Furthermore, $\widetilde{\boldsymbol{\kappa}}(u)$ is isotropic.*

While Figure 3 suggests that more triangles are better, we note that this increases the subspace dimension which makes getting correct projection estimates harder. Instead we choose to stack more meshes with a smaller number of triangles.

Intuitively, since every triangle average depends on many measurements, estimating each average is more robust to measurement corruptions as evidenced in Section 4. Accurate estimates of local averages enable us to recover the geometric structure while being more robust to data errors.

---

[4]We note that this result requires adequate handling of boundary conditions; for the lack of space we omit the straightforward details.

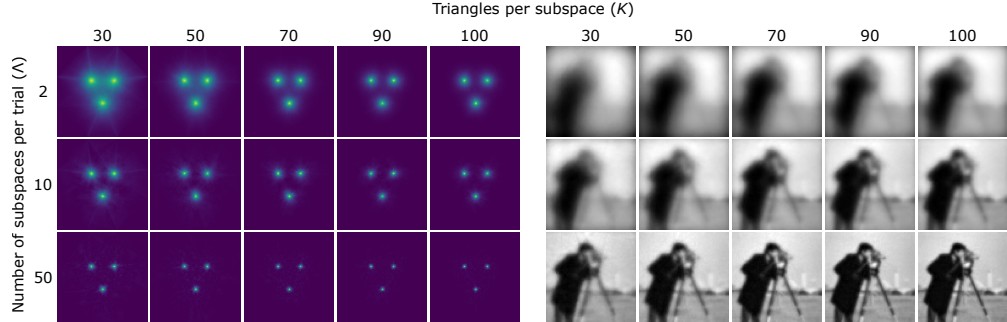

Figure 3: Illustration of the expected kernel $\boldsymbol{\kappa}(u, v)$ with varying subspace dimension, $K$, and number of subspaces, $\Lambda$. Reconstruction of a sparse three-pixel image (left) and the cameraman image (right).

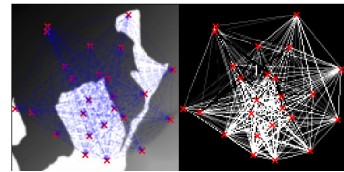

Figure 4: Linearized traveltime tomography illustration: On the left we show a sample model, with red crosses indicating 25 sensor locations and dashed blue lines indicating linearized travel paths; on the right we show a reconstruction from $\binom{25}{2} = 300$ measurements by non-negative least squares.

## 4 NUMERICAL RESULTS

### 4.1 APPLICATION: TRAVELTIME TOMOGRAPHY

To demonstrate our method's benefits we consider linearized traveltime tomography (Hole (1992); Bording et al. (1987)), but we note that the method applies to any inverse problem with scarce data.

In traveltime tomography, we measure $\binom{N}{2}$ wave travel times between $N$ sensors as in Figure 4. Travel times depend on the medium property called slowness (inverse of speed) and the task is to reconstruct the spatial slowness map. Image intensities are a proxy for slowness maps—the lower the image intensity the higher the slowness. In the straight-ray approximation, the problem data is modeled as integral along line segments:

$$y(\boldsymbol{s}_i, \boldsymbol{s}_j) = \int_0^1 x(t\boldsymbol{s}_i + (1-t)\boldsymbol{s}_j) \, \mathrm{d}t, \ \forall \ \boldsymbol{s}_i \neq \boldsymbol{s}_j \tag{3}$$

where $x : \mathbb{R}^2 \to \mathbb{R}^+$ is the continuous slowness map and $\boldsymbol{s}_i, \boldsymbol{s}_j$ are sensor locations. In our experiments, we use a $128 \times 128$ pixel grid with 25 sensors (300 measurements) placed uniformly in an inscribed circle, and corrupt the measurements with zero-mean iid Gaussian noise.

### 4.1.1 ARCHITECTURES AND RECONSTRUCTION

We generate random Delaunay meshes each with 50 triangles. The corresponding projector matrices compute average intensity over triangles to yield a piecewise constant approximation $\boldsymbol{P}_{S_\lambda} \boldsymbol{x}$ of $\boldsymbol{x}$. We test two distinct architectures: (i) **ProjNet**, tasked with estimating the projection into a single subspace; and (ii) **SubNet**, tasked with estimating the projection over multiple subspaces.[5]

The ProjNet architecture is inspired by the FBPConvNet (Jin et al. (2016)) and the U-Net (Ronneberger et al. (2015)) as shown in Figure 11a in the appendix. Crucially, we constrain the network output to live in $S_\lambda$ by fixing the last layer of the network to be a projector, $\boldsymbol{P}_{S_\lambda}$ (Figure 11a). A similar trick in a different context was proposed in (Sønderby et al. (2016)).

---

[5]Code available at `https://github.com/swing-research/deepmesh` under the MIT License.

We combine projection estimates from many ProjNets by regularized linear least-squares (2) to get the reconstructed model (cf. Figure 2) with the regularization parameter $\lambda$ determined on five held-out images. A drawback of this approach is that a separate ProjNet must be trained for each subspace. This motivates the SubNet (shown in Figure 11b). Each input to SubNet is the concatenation of a non-negative least squares reconstruction and 50 basis functions, one for each triangle forming a 51-channel input. This approach scales to any number of subspaces which allows us to get visually smoother reconstructions without any further regularization as in (2). On the other hand, the projections are less precise which can lead to slightly degraded performance.

As a quantitative figure of merit we use the signal-to-noise ratio (SNR). The input SNR is defined as $10\log_{10}(\sigma_{\text{signal}}^2/\sigma_{\text{noise}}^2)$ where $\sigma_{\text{signal}}^2$ and $\sigma_{\text{noise}}^2$ are the signal and noise variance; the output SNR is defined as $\sup_{a,b} 20\log_{10}(\|\boldsymbol{x}\|_2/\|\boldsymbol{x} - a\hat{\boldsymbol{x}} - b\|_2)$ with $\boldsymbol{x}$ the ground truth and $\hat{\boldsymbol{x}}$ the reconstruction.

130 ProjNets are trained for 130 different meshes with measurements at various SNRs. Similarly, a single SubNet is trained with 350 different meshes and the same noise levels. We compare the ProjNet and SubNet reconstructions with a direct U-net baseline convolutional neural network that reconstructs images from their non-negative least squares reconstructions. The direct baseline has the same architecture as SubNet except the input is a single channel non-negative least squares reconstruction like in ProjNet and the output is the target reconstruction. Such an architecture was proposed by (Jin et al. (2016)) and is used as a baseline in recent learning-based inverse problem works (Lunz et al. (2018); Ye et al. (2018)) and is inspiring other architectures for inverse problems (Antholzer et al. (2017)). We pick the best performing baseline network from multiple networks which have a comparable number of trainable parameters to SubNet. We simulate the lack of training data by testing on a dataset that is different than that used for training.

**Robustness to corruption**   To demonstrate that our method is robust against arbitrary assumptions made at training time, we consider two experiments. First, we corrupt the data with zero-mean iid Gaussian noise and reconstruct with networks trained at different input noise levels. In Figures 5a, 12 and Table 1, we summarize the results with reconstructions of geo images taken from the BP2004 dataset[6] and x-ray images of metal castings (Mery et al. (2015)). The direct baseline and SubNet are trained on a set of 20,000 images from the arbitrarily chosen LSUN bridges dataset (Yu et al. (2015)) and tested with the geophysics and x-ray images. ProjNets are trained with 10,000 images from the LSUN dataset. Our method reports better SNRs compared to the baseline. We note that direct reconstruction is unstable when trained on clean and tested on noisy measurements as it often hallucinates details that are artifacts of the training data. For applications in geophysics it is important that our method correctly captures the shape of the cavities unlike the direct inversion which can produce sharp but wrong geometries (see outlines in Figure 5a).

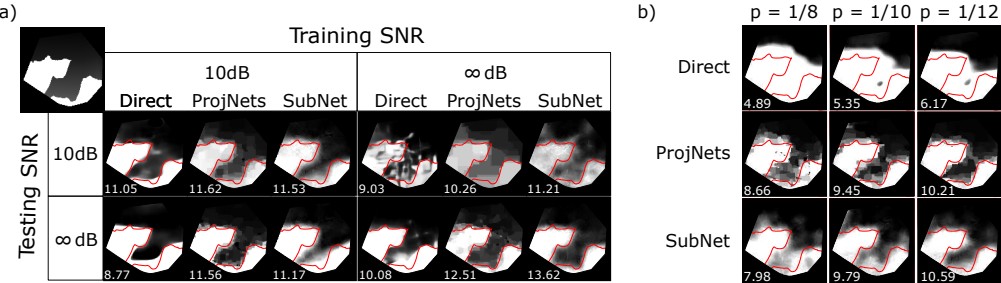

Figure 5: a) Reconstructions for different combinations of training and testing input SNR. The output SNR is indicated for each reconstruction. Our method stands out when the training and testing noise levels do not match; b) reconstructions with erasures with probability $\frac{1}{8}$, $\frac{1}{10}$ and $\frac{1}{12}$. The reconstructions are obtained from networks which are trained with input SNR of 10 dB. The direct network cannot produce a reasonable image in any of the cases.

Second, we consider a different corruption mechanism where traveltime measurements are erased (set to zero) independently with probability $p \in \left\{\frac{1}{12}, \frac{1}{10}, \frac{1}{8}\right\}$, and use networks trained with 10 dB input SNR on the LSUN dataset to reconstruct. Figure 5b and Table 2 summarizes our findings. Unlike

---

[6]http://software.seg.org/datasets/2D/2004_BP_Vel_Benchmark/

| Average SNR over 102 x-ray images | | Training SNR | | | | | |
|---|---|---|---|---|---|---|---|
| | | 10 dB | | | $\infty$ dB | | |
| | | Direct | ProjNets | SubNet | Direct | ProjNets | SubNet |
| Testing SNR | 10 dB | 13.51 | 14.49 | 13.92 | 10.34 | 12.88 | 12.85 |
| | $\infty$ dB | 13.78 | 15.38 | 14.04 | 16.67 | 17.23 | 16.86 |

Table 1: Average reconstruction SNR for various training and testing SNR combinations.

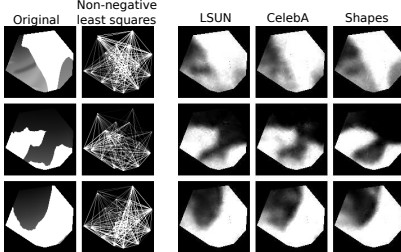

| Average SNR over 102 x-ray images | $p = \frac{1}{8}$ | $p = \frac{1}{10}$ | $p = \frac{1}{12}$ |
|---|---|---|---|
| Direct | 9.03 | 9.62 | 10.06 |
| ProjNets | 11.09 | 11.70 | 12.08 |
| SubNet | 11.33 | 11.74 | 11.99 |

Table 2: Average SNR values for reconstructions from measurements with erasure probability, $p$. All networks were trained for 10dB noisy measurements on the LSUN bridges dataset. Refer to Appendix E for actual reconstructions.

Figure 6: Reconstructions from networks trained on different datasets (LSUN, CelebA and Shapes) with 10dB training SNR.

with Gaussian noise (Figure 5a) the direct method completely fails to recover coarse geometry in all test cases. In our entire test dataset of 102 x-ray images there is not a single example where the direct network captures a geometric feature that our method misses. This demonstrates the strengths of our approach. For more examples of x-ray images please see Appendix E.

**Robustness against dataset overfitting** Figure 6 illustrates the influence of the training data on reconstructions. Training with LSUN, CelebA (Liu et al. (2015)) and a synthetic dataset of random overlapping shapes (see Figure 15 in Appendix for examples) all give comparable reconstructions—a desirable property in applications where real ground truth is unavailable.

We complement our results with reconstructions of checkerboard phantoms (standard resolution tests) and x-rays of metal castings in Figure 7. We note that in addition to better SNR, our method produces more accurate geometry estimates, as per the annotations in the figure.

## 5 CONCLUSION

We proposed a new approach to regularize ill-posed inverse problems in imaging, the key idea being to decompose an unstable inverse mapping into a collection of stable mappings which only estimate

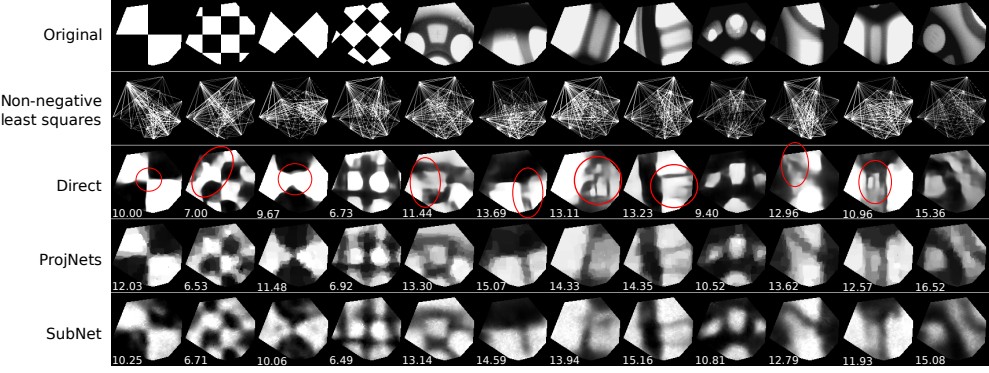

Figure 7: Reconstructions on checkerboards and x-rays with 10dB measurement SNR tested on 10dB trained networks. Red annotations highlight where the direct net fails to reconstruct correct geometry.

low-dimensional projections of the model. By using piecewise-constant Delaunay subspaces, we showed that the projections can indeed be accurately estimated. Combining the projections leads to a deconvolution-like problem. Compared to directly learning the inverse map, our method is more robust against noise and corruptions. We also showed that regularizing via projections allows our method to generalize across training datasets. Our reconstructions are better both quantitatively in terms of SNR and qualitatively in the sense that they estimate correct geometric features even when measurements are corrupted in ways not seen at training time. Future work involves getting precise estimates of Lipschitz constants for various inverse problems, regularizing the reformulated problem using modern regularizers (Ulyanov et al. (2017)), studying extensions to non-linear problems and developing concentration bounds for the equivalent convolution kernel.

## ACKNOWLEDGEMENT

This work utilizes resources supported by the National Science Foundation's Major Research Instrumentation program, grant #1725729, as well as the University of Illinois at Urbana-Champaign.

We gratefully acknowledge the support of NVIDIA Corporation with the donation of one of the GPUs used for this research.

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

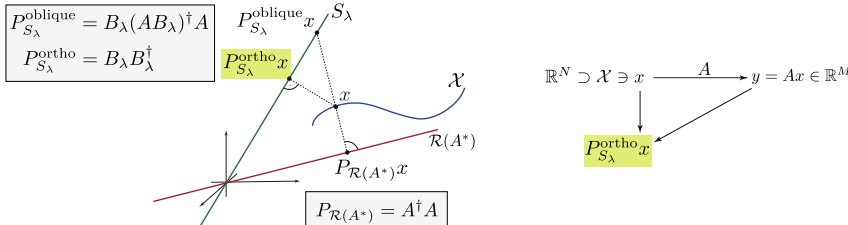

Figure 8: Orthogonal vs. oblique projections. There is no linear operator acting on $\boldsymbol{y}$ or on the orthogonal projection $\widetilde{\boldsymbol{y}} = \boldsymbol{P}_{\mathcal{R}(\boldsymbol{A}^*)}\boldsymbol{x} = \boldsymbol{A}^\dagger\boldsymbol{y}$ that can compute the orthogonal projection into $S$.

## A    NEED FOR NON-LINEAR OPERATORS

We explain the need for non-linear operators even in the absence of noise with reference to Figure 8. Projecting $\boldsymbol{x}$ into a given known subspace is a simple linear operation, so it may not be a priori clear why we use non-linear neural networks to estimate the projections. Alas, we do not know $\boldsymbol{x}$ and only have access to $\boldsymbol{y}$. Suppose that there exists a linear operator (a matrix) $\boldsymbol{F} \in \mathbb{R}^{N \times M}$ which acts on $\boldsymbol{y}$ and computes the projection of $\boldsymbol{x}$ on $S_\lambda$. A natural requirement on $\boldsymbol{F}$ is consistency: if $\boldsymbol{x}$ already lives in $S_\lambda$, then we would like to have $\boldsymbol{F}\boldsymbol{A}\boldsymbol{x} = \boldsymbol{x}$. This implies that for any $\boldsymbol{x}$, not necessarily in $S_\lambda$, we require $\boldsymbol{F}\boldsymbol{A}\boldsymbol{F}\boldsymbol{A}\boldsymbol{x} = \boldsymbol{F}\boldsymbol{A}\boldsymbol{x}$ which implies that $\boldsymbol{F}\boldsymbol{A} = (\boldsymbol{F}\boldsymbol{A})^2$ is an idempotent operator. Letting the columns of $\boldsymbol{B}_\lambda$ be a basis for $S_\lambda$, it is easy to see that the least squares minimizer for $\boldsymbol{F}$ is $\boldsymbol{B}_\lambda(\boldsymbol{A}\boldsymbol{B}_\lambda)^\dagger$. However, because $\mathcal{R}(\boldsymbol{F}) = S_\lambda \neq \mathcal{R}(\boldsymbol{A}^*)$ ($\boldsymbol{A}^*$ is the adjoint of $\boldsymbol{A}$, simply a transpose for real matrices), in general it will not hold that $(\boldsymbol{F}\boldsymbol{A})^* = \boldsymbol{F}\boldsymbol{A}$. Thus, $\boldsymbol{F}\boldsymbol{A}$ is an oblique, rather than orthogonal projection into $S$. In Figure 8 this corresponds to the point $\boldsymbol{P}_{S_\lambda}^{\text{oblique}}\boldsymbol{x}$ which can be arbitrarily far from the orthogonal projection $\boldsymbol{P}_{S_\lambda}^{\text{ortho}}\boldsymbol{x}$. The nullspace of the oblique projection is precisely $\mathcal{N}(\boldsymbol{A}) = \mathcal{R}(\boldsymbol{A}^*)^\perp$.

Thus consistent linear operators can at best yield oblique projections which can be far from the orthogonal one. One could also see this geometrically from Figure 8. As the angle between $S_\lambda$ and $\mathcal{R}(\boldsymbol{A}^*)$ increases to $\pi/2$ the oblique projection point travels to infinity (note that the oblique projection always happens along the nullspace of $\boldsymbol{A}$, which is the line orthogonal to $\boldsymbol{P}_{\mathcal{R}(\boldsymbol{A}^*)}$. Since our subspaces are chosen at random, in general they are not aligned with $\mathcal{R}(\boldsymbol{A}^*)$. The only subspace on which we can linearly compute an orthogonal projection from $\boldsymbol{y}$ is $\mathcal{R}(\boldsymbol{A}^*)$; this is given by the Moore-Penrose pseudoinverse. Therefore, to get the orthogonal projection onto random subspaces, we must use non-linear operators. More generally, for any other ad hoc linear reconstruction operator $\boldsymbol{W}$, $\boldsymbol{W}\boldsymbol{y} = \boldsymbol{W}\boldsymbol{A}\boldsymbol{x}$ always lives in the column space of $\boldsymbol{W}\boldsymbol{A}$ which is a subspace whose dimension is at most the number of rows of $\boldsymbol{A}$. However, we do not have any linear subspace model for $\boldsymbol{x}$.

As shown in the right half of Figure 8, as soon as $\boldsymbol{A}$ is injective on $\mathcal{X}$, the existence of this non-linear map is guaranteed by construction: since $\boldsymbol{y}$ determines $\boldsymbol{x}$, it also determines $\boldsymbol{P}_{S_\lambda}\boldsymbol{x}$.

We show the results of numerical experiments in Figures 9 and 10 which further illustrate the performance difference between linear oblique projectors and our non-linear learned operator when estimating the projection of an image into a random subspace. We refer the reader to the captions below each figure for more details.

## B    PROOF OF PROPOSITION 1

*Proof.* The reconstruction of the new inverse problem can be written as $\widehat{\boldsymbol{x}} = \widetilde{\boldsymbol{B}}\boldsymbol{B}^\top\boldsymbol{x}$ where the columns of $\widetilde{\boldsymbol{B}} = (\boldsymbol{B}^\top)^\dagger$ form a biorthogonal basis to the columns of $\boldsymbol{B}$. Thus

$$\widehat{\boldsymbol{x}} = \sum_{p=1}^{K\Lambda} \langle \boldsymbol{x}, \boldsymbol{b}_p \rangle \, \widetilde{\boldsymbol{b}}_p.$$

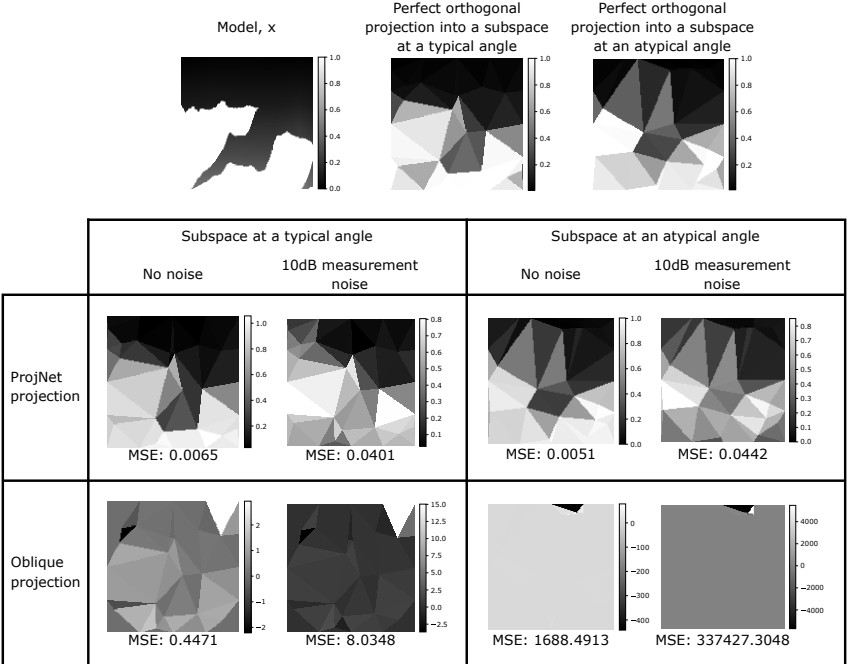

Figure 9: Comparison between perfect orthogonal projection, ProjNet projections and oblique projection. The projections of an image, $x$, (same as Figure 5) are obtained using ProjNet and the linear oblique projection method. The mean-squared errors (MSE) between the obtained projections and the perfect projections are stated. The subspaces used in this figure were used in the ProjNet reconstructions.

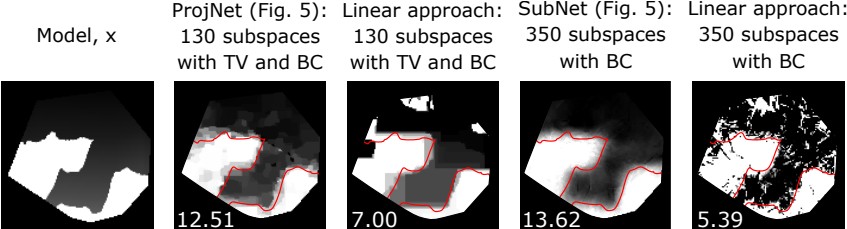

Figure 10: We try hard to get the best reconstruction from the linear approach. SNRs are indicated in the bottom-left of each reconstruction. In the linear approach, coefficients are obtained using the linear oblique projection method. Once coefficients are obtained, they are non-linearly reconstructed according to (2). Both linear approach reconstructions use the box-constraint (BC) mentioned in (2). For the 130 subspace reconstruction total-variation (TV) regularization is also used. Therefore, once the coefficients are obtained using the linear approach, the reconstruction of the final image is done in an identical manner as ProjNet for 130 subspaces and SubNet for 350 subspaces. To give the linear approach the best chance we also optimized hyperparameters such as the regularization parameter to give the highest SNR.

Using the definition of the inner product and rearranging, we get

$$\widehat{\boldsymbol{x}}(u) = \left\langle \sum_{p=1}^{K\Lambda} \boldsymbol{b}_p(\cdot)\widetilde{\boldsymbol{b}}_p(u),\ \boldsymbol{x} \right\rangle \stackrel{\text{def}}{=} \langle \boldsymbol{\kappa}(u,\cdot), \boldsymbol{x} \rangle$$

where $\boldsymbol{\kappa}(u,v) \stackrel{\text{def}}{=} \sum_{p=1}^{K\Lambda} \boldsymbol{b}_p(v)\widetilde{\boldsymbol{b}}_p(u)$. Now, the probability distribution of triangles around any point $u$ is both shift- and rotation-invariant because a Poisson process in the plane is shift- and rotation-invariant. It follows that $\mathbb{E}\,\boldsymbol{\kappa}(u,v) = \widetilde{\boldsymbol{\kappa}}(\|u-v\|)$ for some $\widetilde{\boldsymbol{\kappa}}$, meaning that

$$(\mathbb{E}\,\widehat{\boldsymbol{x}})(u) = \mathbb{E}\,\langle \boldsymbol{\kappa}(u,\ \cdot), \boldsymbol{x} \rangle = \langle \widetilde{\boldsymbol{\kappa}}(\|u-\cdot\|),\ \boldsymbol{x} \rangle = (\boldsymbol{x} * \widetilde{\boldsymbol{\kappa}})(u)$$

which is a convolution of the original model with a rotationally invariant (isotropic) kernel. □

## C  NETWORK ARCHITECTURES

Figure 11 explains the network architecture used for ProjNet and SubNet. The network consists of a sequence of downsampling layers followed by upsampling layers, with skip connections (He et al. (2016b;a)) between the downsampling and upsampling layers. Each ProjNet output is constrained to a single subspace by applying a subspace projection operator, $\boldsymbol{P}_{S_\lambda}$. We train 130 such networks and reconstruct from the projection estimate using (2). SubNet is a single network that is trained over multiple subspaces. To do this, we change its input to be $[\widetilde{\boldsymbol{y}}\ \boldsymbol{B}_\lambda]$. Moreover, we apply the same projection operator as ProjNet to the output of the SubNet. Each SubNet is trained to give projection estimates over 350 random subspaces. This approach allows us to scale to any number of subspaces without training new networks for each. Moreover, this allows us to build an over-constrained system $\boldsymbol{q} = \boldsymbol{Bx}$ to solve. Even though SubNet has almost as many parameters as the direct net, reconstructing via the projection estimates allows SubNet to get higher SNR and more importantly, get better estimates of the coarse geometry than the direct inversion. All networks are trained with the Adam optimizer.

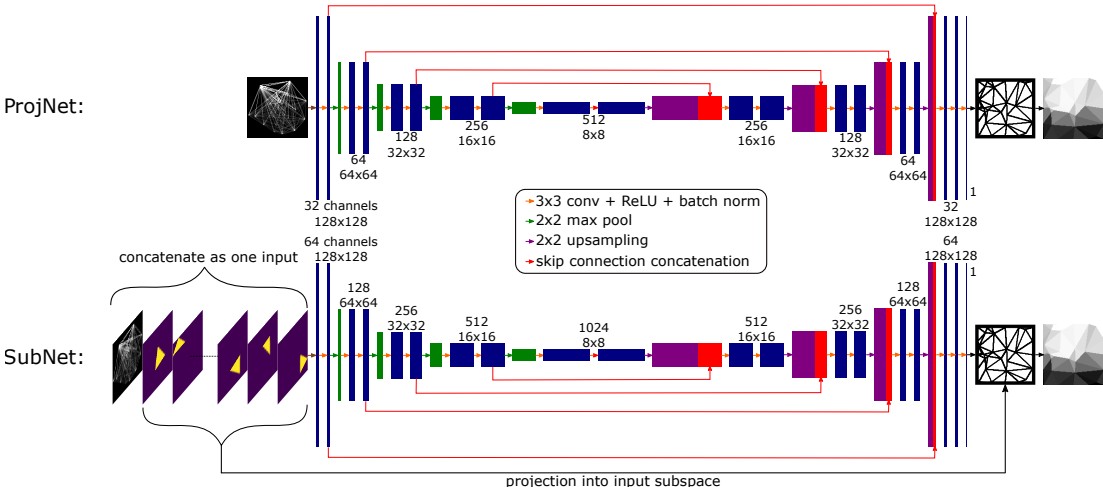

Figure 11: a) ProjNet architecture; b) SubNet architecture. In both cases, the input is a non-negative least squares reconstruction and the network is trained to reconstruct a projection into one subspace. In SubNet, the subspace basis is concatenated to the non-negative least squares reconstruction.

## D  FURTHER RECONSTRUCTIONS

We showcase more reconstructions on actual geophysics images taken from the BP2004 dataset in Figure 12. Note that all networks were trained on the LSUN bridges dataset.

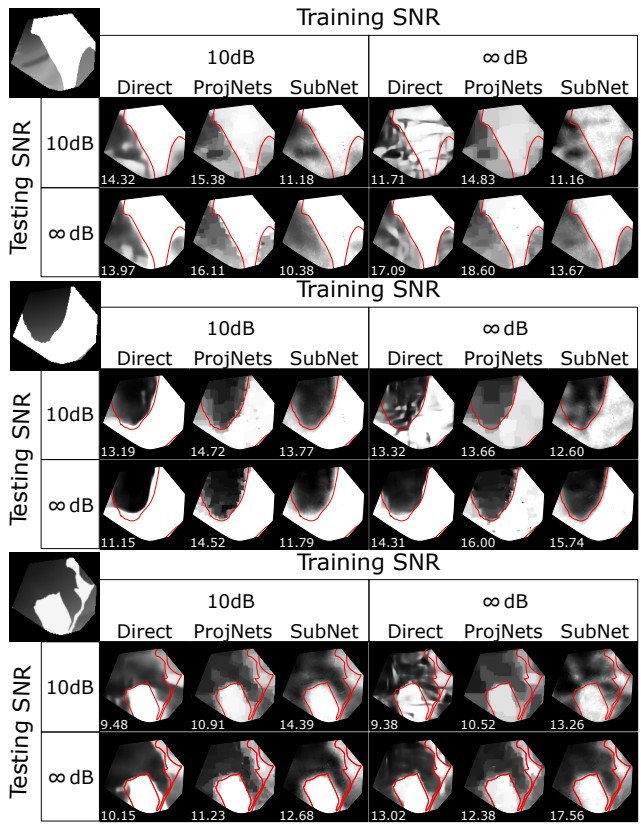

Figure 12: Geophysics image patches taken from BP2004 dataset. Our method especially gets correct global shapes with better accuracy even when tested on noise levels different from training.

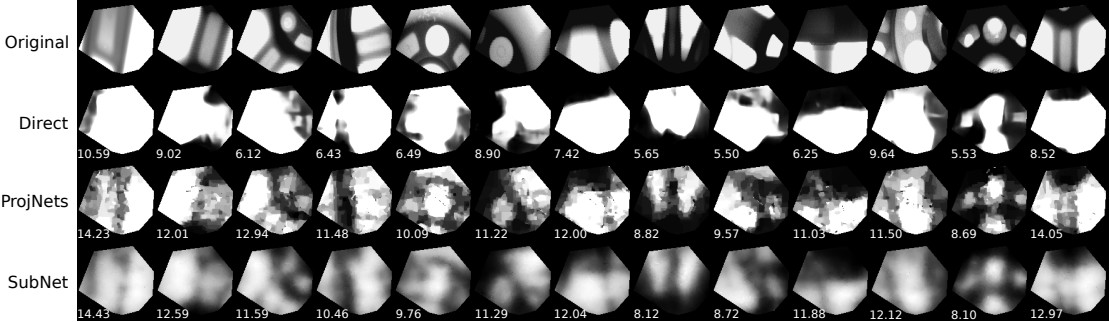

Figure 13: Reconstructions from erasures on x-ray images with erasure probability $p = \frac{1}{8}$.

## E ERASURE RECONSTRUCTIONS

We show additional reconstructions for the largest corruption case, $p = \frac{1}{8}$, for x-ray images (Figure 13) and geo images (Figure 14). Our method consistently has better SNR. More importantly we note that there is not a single instance where the direct reconstruction gets a feature that our methods do not. In a majority of instances, the direct network misses a feature of the image. This is highly undesirable in settings such as geophysical imaging.

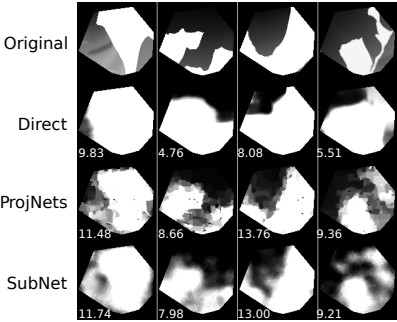

Figure 14: Reconstructions from erasures on geo images with erasure probability $p = \frac{1}{8}$.

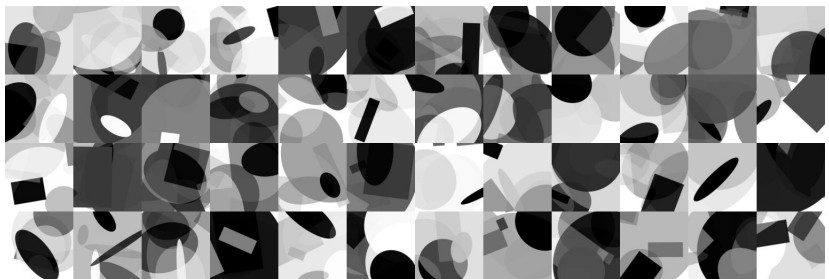

Figure 15: Examples from the random shapes dataset which is used in Figure 6.

## F  SHAPES DATASET

The shapes dataset was generated using random ellipses, circle and rectangle patches. See Figure 15 for examples. This dataset was used in Figure 6.

## G  COMPARISON OF PROPOSED METHOD WITH ENSEMBLE OF DIRECT NETS

In Section 4 we train multiple ProjNets, each focusing on a different low-dimensional subspace. Here we train an ensemble of direct networks where each network is as described in Section 4.1.1 and evaluate the robustness of a method where the outputs of these networks are averaged to give a final reconstruction. Once again, we consider scenarios where the model is trained with data at a particular noise level and then tested with data at a different noise level and with erasures that were unseen during training time. We show that our proposed method is more robust to changes in the test scenario.

In Figure 16, we consider the erasure model with $p = \frac{1}{8}$ (described in Figure 5b). 9 out of 10 randomly chosen direct network reconstructions fail to capture the key structure of the original image under this corruption mechanism. In Table 3, we summarize this with the SNRs of reconstructions from the erasure corruption mechanism. In that table we also report SNRs when reconstructing from measurements at different noise levels. The ensemble of direct networks performs well when the training and test data have the same measurement noise level. However, our method is more robust to changes in the test noise level. This further illustrates that direct networks are highly tuned to the training scenario and therefore not as stable as our proposed method (cf. Section 3).

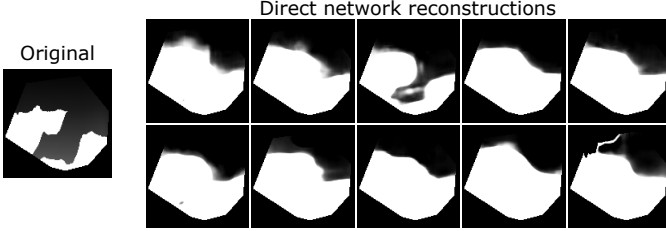

Figure 16: Reconstructions of the original image from 10 individually trained direct inversion networks for 10dB noise under the $p = \frac{1}{8}$ erasure corruptions model (described in Figure 5b). 9 out of the 10 reconstructions fail to capture the key structure of the original image.

| Scenario | Single Direct | 50-Ensemble | ProjNets | SubNet |
|---|---|---|---|---|
| 10 dB train and 10 db test | 12.01 | 13.404 | 13.15 | 12.72 |
| 10 dB train and $\infty$ dB test | 11.01 | 10.70 | 13.36 | 11.51 |
| 10 dB train and erasures with $p = \frac{1}{8}$ test | 7.19 | 8.70 | 10.64 | 10.48 |

Table 3: Comparison of SNRs in different scenarios. 50-Ensemble refers to reconstructions obtained by averaging the output of 50 direct networks.

