# OpenReview forum: "Random mesh projectors for inverse problems"
_ICLR.cc/2019/Conference_

### Official Review · AnonReviewer3 · 2018-11-02
**Unclear why this should work**

**Rating:** 4
**Confidence:** 3

**Review:**

This paper describes a novel method for solving inverse problems in imaging.

The basic idea of this approach is use the following steps:
1. initialize with nonnegative least squares solution to inverse problem (x0)
2. compute m different projections of x0
3. estimate x from the m different projections by solving "reformuated" inverse problem using TV regularization.

The learning part of this algorithm is in step 2, where m different convolutional neural networks are used to learn m good projections. The projections correspond to computing a random Delaunay triangulation over the image domain and then computing pixel averages within each triangle. It's not clear exactly what the learning part is doing, i.e. what makes a "good" triangulation, why a CNN might accurately represent one, and what the shortcomings of truly random triangulations might be.

More specifically, for each projection the authors start with a random set of points in the image domain and compute a Delaunay triangulation. They average x0 in each of the Delaunay triangles. Then since the projection is constant on each triangle, the projection into the lower-dimensional space is given by the magnitude of the function over each of the triangular regions. Next they train a convolutional neural network to approximate the above projection. The do this m times. It's not clear why the neural network approximation is necessary or helpful.

Empirically, this method outperforms a straightforward use of a convolutional U-Net to invert the problem.

The core novelty of this paper is the portion that uses a neural network to calculate a projection onto a random Delaunay triangulation. The idea of reconstructing images using random projections is not especially new, and much of the "inverse-ness" of the problem here is removed by first taking the pseudoinverse of the forward operator and applying it to the observations. Then the core idea at the heart of the paper is to speed up this reconstruction using a neural network by viewing the projection onto the mesh space as a set of special filter banks which can be learned.

At the heart of this paper is the idea that for an L-Lipschitz function f : R^k → R the sample complexity
is O(L^k), so the authors want to use the random projections to essentially reduce L. However, the Cooper sample complexity bound scales with k like k^{1+k/2}, so the focus on the Lipschitz constant seems misguided.
This isn't damning, but it seems like the piecewise-constant estimators are a sort of regularizer, and that's where we
really get the benefits.

The authors only compare to another U-Net, and it's not entirely clear how they even trained that U-Net. It'd be nice to see if you get any benefit here from their method relative to other approaches in the literature, or if this is just better than inversion using a U-Net. Even how well a pseudoinverse does would be nice to see or TV-regularized least squares.

Practically I'm quite concerned about their method requiring training 130 separate convolutional neural
nets. The fact that all the different datasets give equal quality triangulations seems a bit odd, too. Is
it possible that any network at all would be okay? Can we just reconstruct the image from regression
on 130 randomly-initialized convolutional networks?

The proposed method isn't bad, and the idea is interesting. But I can't help but wonder whether it works just because what we're doing is denoising the least squares reconstruction, and regression on many random projections might be pretty good for that. Unfortunately, the experiments don't help with developing a deeper understanding.

---

> ### Author Response · Authors · 2018-11-16
> **Clarification of our method (Part 1)**
>
> The reviewer summarizes our method as
>
> >> “This paper describes a novel method for solving inverse problems in imaging.
>
> The basic idea of this approach is use the following steps:
> 1. initialize with nonnegative least squares solution to inverse problem (x0)
> 2. compute m different projections of x0
> 3. estimate x from the m different projections by solving "reformuated" inverse problem using TV regularization.”
>
> Response: We have to respectfully disagree with this summary, especially because it informs the remainder of the reviewer’s comments. There seems to be a misunderstanding about Step 2 and many later comments appear to stem from it. Since this step is the crux of our proposed method, we begin by summarizing it here, with references to the relevant parts of the manuscript.
>
> Instead of computing m different projections of x0 as the reviewer suggests, we regress subspace projections of x, the true image (see Section 3.1.1, Paragraphs 3 and 4). To do so, we must train a nonlinear regressor, in our case a convolutional neural network. (The need for nonlinearity is explained below.) To make this point clearer in the manuscript, we updated Figure 2 to explicitly show that x0 is not fed into linear subspace projectors of itself, but rather used as data from which we estimate projections of x. Indeed, projecting x0 would not be very interesting since it would simply imply various linear ways of looking at x0 and the networks would not be doing any actual inversion or data modeling.
>
> Again, what we actually do is that we compute *orthogonal* projections P_S x from y = Ax (or x0 = pinv(A)y or something similar) into a collection of subspaces {S_\lambda}_{\lambda=1}^{\Lambda} (see Section 3.1.1, Paragraph 3). While projecting x0 is a simple linear operation, regressing projections of an unknown x from the measurement data y is not. To explain why we need nonlinear regressors, we added a new figure and a short discussion to the manuscript (please see the new Appendix A). For the reviewer’s convenience, we summarize the discussion here (although it might be easier to read in the typeset pdf version):
>
> Suppose that there exists a linear operator F \in R^{N \times M} which maps y (or pinv(A)y) to P_S x. The simplest requirement on such an F is consistency: if x already lives in the subspace S, then we would like to have F A x = x. Another way to write this is that for any x, not necessarily in S, we require FA FA x = FA x, which implies that FA = (FA)^2 is an idempotent operator. However, because range(F) = S \neq range(A^*), it will in general not hold that (FA)^* = FA. This implies that FA is not an orthogonal projection, but rather an oblique one.
>
> As we show in the new Figure 8 (Appendix A), this oblique projection can be an arbitrarily poor approximation of the actual orthogonal projection that we seek. The nullspace of this projection is precisely N(A) = range^\perp(A^*). Similar conclusions can be drawn for any other (ad hoc) linear operator, which would not even be a projection.
>
> There are various assumptions one can make to guarantee that the map from Ax to P_S x exists. We assume that the models live on a low-dimensional manifold (please see updated Section 3.1; this low-dimensional structure assumption has previously been a footnote), and that the measurements are in general position with respect to this manifold. Our future work involves making quantitative statements about this aspect of the method.

---

> > ### Author Response · Authors · 2018-11-16
> > **Clarification of our method (Part 2)**
> >
> > >> “The learning part of this algorithm is in step 2, where m different convolutional neural networks are used to learn m good projections. The projections correspond to computing a random Delaunay triangulation over the image domain and then computing pixel averages within each triangle. It's not clear exactly what the learning part is doing, i.e. what makes a "good" triangulation, why a CNN might accurately represent one, and what the shortcomings of truly random triangulations might be.”
> >
> > Response: Again, we feel that there is a misunderstanding about the role of the networks in our method. In one of the subsequent comments the reviewer posits that
> >
> > >> “... the core idea at the heart of the paper is to speed up this reconstruction using a neural network by viewing the projection onto the mesh space as a set of special filter banks which can be learned.”
> >
> > It seems that the reviewer’s interpretation is that the triangulation is computed by the network, together with the projection. But as we elaborate in Section 3.1.1, and as we explained when commenting on the reviewer’s summary above, the role of the network is to compute an orthogonal projection of x into a given random subspace. This is ensured by an explicit, non-trainable, projector added as the last layer (see Section 4.1.1, Paragraph 2).  As such, we do use *truly random triangulations*. Nothing about these triangulations is being learned from the training data (see Section 3.1.1, Paragraph 3).
> >
> > As detailed in the discussion above, computing these projections from y (or x0) is a nonlinear problem, and it requires a nonlinear computational structure (please see the new Appendix A). Since the normal operator corresponding to ray transforms is convolutional, we decided to use a CNN (assuming that our rays provide a somewhat constant coverage of the domain). As the reviewer points out, the CNNs are not natural structures to produce images that live exactly on triangulations. That is why we add an explicit, non-trainable projection layer (see Section 4.1.1, Paragraph 2, and Figure 11).
> >
> > >> “More specifically, for each projection the authors start with a random set of points in the image domain and compute a Delaunay triangulation. They average x0 in each of the Delaunay triangles. Then since the projection is constant on each triangle, the projection into the lower-dimensional space is given by the magnitude of the function over each of the triangular regions. Next they train a convolutional neural network to approximate the above projection. The do this m times. It's not clear why the neural network approximation is necessary or helpful.”
> >
> > Response: We again wish to emphasize that it is not x0 that we project (this is just a linear operation), but rather we nonlinearly regress orthogonal low-dimensional projections of x which implies that the network models some aspects of the distribution of x. We agree with the reviewer that in the former case, the network would be superfluous.
> >
> > >> “The core novelty of this paper is the portion that uses a neural network to calculate a projection onto a random Delaunay triangulation. The idea of reconstructing images using random projections is not especially new, and much of the "inverse-ness" of the problem here is removed by first taking the pseudoinverse of the forward operator and applying it to the observations. Then the core idea at the heart of the paper is to speed up this reconstruction using a neural network by viewing the projection onto the mesh space as a set of special filter banks which can be learned.”
> >
> > Response: While we agree with the reviewer that random projections are a known idea, as far as we know and as noted by Reviewer 2, this is the first work that attempts to regress the orthogonal projections of the target signal x into random subspaces. We believe that this contribution sets it apart from previous work, especially because computing these projections from measurements is a truly nonlinear problem unlike the more common fixed linear projections. The reason to regress P_S x instead of x is that it is a more stable task, and a “clever” way to achieve randomization while at the same time controlling stability and hardness of learning. The role of the network is to approximate this nonlinear operator that maps y to projections of x, rather than to speed up a simple linear projection of x0.
> >
> > We also respectfully disagree that much of the inverseness is removed by taking the pseudoinverse. In fact, this is one of our main contributions: we state in several places in the manuscript (for example Paragraph 3 of Introduction), that we work in a highly undersampled regime where the pseudoinverse (or any other simple regularizer for that matter) cannot do a reasonable job and the role of learning cannot be seen as denoising or artifact removal (see for example Figure 1 bottom row). This is also illustrated in  Section 4 with the non-negative least squares reconstructions shown in Figures 6 and 7.

---

> > > ### Author Response · Authors · 2018-11-16
> > > **Clarification of our method (Part 3)**
> > >
> > > >> “At the heart of this paper is the idea that for an L-Lipschitz function f : R^k → R the sample complexity is O(L^k), so the authors want to use the random projections to essentially reduce L. However, the Cooper sample complexity bound scales with k like k^{1+k/2}, so the focus on the Lipschitz constant seems misguided. This isn't damning, but it seems like the piecewise-constant estimators are a sort of regularizer, and that's where we really get the benefits.”
> > >
> > > Response: We apologize for using K in the manuscript while stating this result: this is unfortunate, especially because we later use K for subspace dimension (and in this case the reviewer is absolutely right). We are interested in the stability of the map from measurements y to the targets P_S x, so that the map f(y) operates on objects in R^M. Note that the number of measurements M (or k in the reviewer’s comment) is kept fixed. On the other hand, L changes because we are learning a simpler target.
> > >
> > > We agree that the piecewise-constant estimators act as a regularizer in the sense of learning. They restrict the hypothesis class to “regular” or “simple” maps, and one standard way to quantify regularity is via the Lipschitz constant.
> > >
> > > >> “The authors only compare to another U-Net, and it's not entirely clear how they even trained that U-Net. It'd be nice to see if you get any benefit here from their method relative to other approaches in the literature, or if this is just better than inversion using a U-Net. Even how well a pseudoinverse does would be nice to see or TV-regularized least squares.”
> > >
> > > Response: We describe the training of the U-Net (the direct baseline in the paper) in some detail in Section 4.1.1, which we now expanded to include information about the number of samples for training. We also now explicitly highlight that the training and test sets are entirely different in all experiments and for all networks. The U-Net that we use achieves state of the art results on a very long list of image recovery tasks [1,2,3,4], including tomographic problems that are similar to the one we experiment with. This suggests that it is a hard baseline to beat. Indeed, as the reviewer suggests, we already do show both the pseudoinverse in Figures 1, 6 and 7 and the TV-regularized least squares in bottom row of Figure 1. It can be observed that in all cases the U-Net, our baseline, outperforms them while the proposed method beats the U-Net.
> > >
> > > >> “Practically I'm quite concerned about their method requiring training 130 separate convolutional neural nets. The fact that all the different datasets give equal quality triangulations seems a bit odd, too. Is it possible that any network at all would be okay? Can we just reconstruct the image from regression on 130 randomly-initialized convolutional networks?”
> > >
> > > Response: We agree that it is favorable to train fewer networks. However, we already do propose the SubNet (motivated exactly by this concern) which requires training only a single network (see Section 4.1.1 Paragraph 5), and which performs on par with the collection of ProjNets and better than the baseline. Note that we are using the same number of samples to train the SubNet and the direct baseline, and only half of those samples to train *all* the ProjNets. We now mention the number of samples explicitly in Section 4 under Robustness to Corruption.
> > >
> > > We are not quite certain that we understand the comment about equal quality triangulations. The experiments on different datasets showcase that we can train on arbitrary image datasets and obtain comparable reconstructions. We reiterate that our networks are not computing triangulations, only projections into these triangular subspaces. All triangulations are generated at random, independently of the datasets and the networks.
> > >
> > > The reviewer’s idea of regression on 130 randomly-initialized convolutional networks is interesting and a possible avenue for further research. However, each network would approximate the same unstable, high variance map (see, for example, the response to Reviewer 2, and examples https://tinyurl.com/direct-new-seeds ). One important aspect of our randomization via random triangulations is that it gives interpretable, local measurements, equivalent to a new forward operator B with favorable properties (see the discussion in Section 3.2 and 3.3). It is not immediately clear how one would interpret the outputs of randomly initialized convolutional networks.

---

> > > > ### Author Response · Authors · 2018-11-16
> > > > **Clarification of our method (Part 4)**
> > > >
> > > > >> “The proposed method isn't bad, and the idea is interesting. But I can't help but wonder whether it works just because what we're doing is denoising the least squares reconstruction, and regression on many random projections might be pretty good for that. Unfortunately, the experiments don't help with developing a deeper understanding.”
> > > >
> > > > Response: As we stress in the manuscript (Paragraph 3 of Introduction and Figure 1) we are precisely addressing the regime where the denoising or artifact removal paradigm fails. In Figure 1, we show that standard methods that would indeed correspond to denoising the least squares reconstruction, such as the TV-regularized least squares or non-negative least squares do not give a reasonable solution to our problem.
> > > >
> > > > We feel the reviewer’s impression is based on their interpretation that we project x0 into random subspaces, but as we try to emphasize in our response, we are doing something very different. Estimating *orthogonal* projections of x (as opposed to x0) from few measurements cannot be interpreted as denoising, but rather as discovering different stable pieces of information about the conditional distribution of x which is supported on some a priori unknown low-dimensional structure, $\mathcal{X}$, and the part of learning is to discover this structure (or rather, its projections into a set of random subspaces which is a simpler problem). We updated the manuscript to further emphasize this aspect in Section 3.1 and added Appendix A.
> > > >
> > > > [1] Jin, K.H., McCann, M.T., Froustey, E. and Unser, M., 2017. Deep convolutional neural network for inverse problems in imaging. IEEE Transactions on Image Processing, 26(9), pp.4509-4522.
> > > > [2] Rivenson, Y., Zhang, Y., Günaydın, H., Teng, D. and Ozcan, A., 2018. Phase recovery and holographic image reconstruction using deep learning in neural networks. Light: Science & Applications, 7(2), p.17141.
> > > > [3] Sinha, A., Lee, J., Li, S. and Barbastathis, G., 2017. Lensless computational imaging through deep learning. Optica, 4(9), pp.1117-1125.
> > > > [4] Li, S., Deng, M., Lee, J., Sinha, A. and Barbastathis, G., 2018. Imaging through glass diffusers using densely connected convolutional networks. Optica, 5(7), pp.803-813.

---

> > ### Comment · AnonReviewer3 · 2018-11-20
> > **clarification was helpful**
> >
> > Thank you for the clarification on how the method works; this cleared up some things. However, it's still not clear to me why this should work. I agree with the other reviewer that the ensemble hypothesis is one potential explanation, but the paper would be strengthened by more depth in this regard.
> >
> > It would also help to see some concreteness to some of the explanations. I find Appendix A and Figure 8 difficult to follow. Is \cal R(A*) the range of the adjoint of A? I can't find this defined anywhere. Likewise, I can't find a concrete definition of P_S^oblique.
> >
> > Consider this comparison point. Let S_k be a random subspace and U_k be a basis spanning it. Then z_k := P_{S_k} x = U_k v_k for some coefficient vector v_k. Thus one estimator of z_k is simply \hat z_k = U_k \hat v_k where \hat v_k = pinv(AU_k)y. From these \hat z_k's I could then estimate x via
> > \min x \sum_k \|P_{S_k} x - \hat z_k \|_F
> >
> > I think the above approach is consistent with the spirit of your method, but based on linear estimators of z_k instead of CNNs. But this raises several questions:
> >
> > 1. Is the \hat z_k above (which I think might correspond to your P_S^oblique) consistent with your appendix A claim that the oblique projection can be arbitrarily bad? I find this difficult to interpret. If I observe only a subset of entries in a vector that lies in a known subspace, under some conditions I can identify the original location in the subspace. This fact is at the heart of low-rank matrix completion and it seems to contradict your claim about how difficult to can be to compute these projections. How do I interpret your claim in this setting?
> >
> > 2. What is wrong with my approach? Is there an example where it would fail spectacularly but your method would work? Why? How does it compare empirically to the proposed approach? In other words, within this general framework, what is the benefit of nonlinear estimates of the z_k's?
> >
> > 3. In my (admittedly possibly suboptimal) linear approach, do we have any insight into the role of the different orthogonal projections and how performance scales with the number of projections? Perhaps this could provide insight into how the nonlinear version works.
> >
> > 4. What is the role of TV regularization in the final estimation of x? I thought that the different subspace projections were providing a form of regularization, so I was surprised that additional regularization was required.

---

> > > ### Author Response · Authors · 2018-11-25
> > > **Nonlinearity is required (Part 4)**
> > >
> > > >> “2. What is wrong with my approach? Is there an example where it would fail spectacularly but your method would work? Why? How does it compare empirically to the proposed approach? In other words, within this general framework, what is the benefit of nonlinear estimates of the z_k's?”
> > >
> > > Response: Per all the discussions above, your approach will indeed fail spectacularly whenever the random subspaces are not well-aligned with R(A^*). It will work for those that are well-aligned, but in these cases it will not be much more informative than a variation on the pseudoinverse. To further support this claim, we ran numerical experiments to simulate your proposed approach. We have added results from these experiments in the manuscript. If this is too big a change, we are happy to remove it. We provide a link to the results ( https://tinyurl.com/obliqueandprojnetfigure ) and code ( https://tinyurl.com/obliqueandprojnetcode ) if you would like to experiment yourself. Again, note that the subspaces we use are the ones we used in the experiments in the manuscript.
> > >
> > > Again, the benefit of nonlinear estimates of the z_k’s is that they can exploit nonlinear correlations between the nullspace and the R(A^*) components of x, while linear estimators cannot. That is why our reconstructions in the new examples are much better than the linear ones.
> > >
> > > >> “3. In my (admittedly possibly suboptimal) linear approach, do we have any insight into the role of the different orthogonal projections and how performance scales with the number of projections? Perhaps this could provide insight into how the nonlinear version works.”
> > >
> > > Response: Unfortunately, as we argue in this response (and empirical results provided), the linear method provides little insight into what the non-linear method is doing (beyond pointing to the need for nonlinearity). Since it cannot exploit interesting signal models, everything is dictated by the nullspace of A.
> > >
> > > We agree with the reviewer that our approach opens up questions and research opportunities beyond the current manuscript and that various parts merit deeper study. We are excited by this research and intend to write about it in due time. But we also feel that the manuscript proposes a new, useful approach to regularization, that the discussions therein (strengthened by the previous and this round of reviewer’s comments) motivate the method well and provide mathematical intuitions for why the method does work. The 8-page manuscript and the appendix are tightly packed with problem description, mathematical motivations, intuitions, and numerical examples which show that the method outperforms strong baselines. It now has additional discussions about oblique vs orthogonal projections and the need for nonlinearity, and some additional numerical results in the appendices motivated by the reviewer’s concerns. Everything will be backed up by reproducible code (it already is but we cannot publish it due to anonymity). It would thus be very challenging to add significant new material to the current draft. We hope that the reviewer finds our explanations and this statement reasonable.
> > >
> > > >> “4. What is the role of TV regularization in the final estimation of x? I thought that the different subspace projections were providing a form of regularization, so I was surprised that additional regularization was required.”
> > >
> > > As we discuss in the manuscript, the TV-norm regularization is not essential. In fact, for our SubNet (single network that estimates all subspace projections) reconstructions we do not use any regularization, as we already state in the experimental section of the manuscript (Section 4.1.1 Paragraph 3). We make this more explicit by adding a sentence when discussing Equation 2 in Section 3.2. Please note that in the original problem TV regularization did not give workable reconstructions (see Introduction, Figure 1 bottom row and Part 2 of the response to your initial review) so it is an example of how the reformulated inverse problem is better behaved. We use TV regularization for ProjNet reconstructions because we have coefficients for fewer subspaces (130 vs 350) than for SubNet which makes the problem slightly underdetermined. It is not essential for the method to work and even without it it outperforms the baseline as evidenced by SubNet reconstructions, but it does point to the possibility of using more sophisticated strategies in Stage 2, as noted by Reviewer 1.

---

> > > ### Author Response · Authors · 2018-11-25
> > > **Nonlinearity is required (Part 3)**
> > >
> > > >> “If I observe only a subset of entries in a vector that lies in a known subspace, under some conditions I can identify the original location in the subspace.”
> > >
> > > Response: That is certainly true, and something one could use when x lives in a known subspace. But our x does not live in a subspace (let alone a known one). Again, most interesting classes of images (natural, biomedical, seismic, textures) are not well-modeled by subspaces but rather by sparse models, manifold models, etc.
> > >
> > > Moreover, often only a class of models is known: we assume that x lives on a low-dimensional manifold, but we do not know which one (so we have to learn it implicitly from the data). A different example is that we might know that x is sparse in a dictionary, but without knowing the dictionary we need to learn it.
> > >
> > > Even if we simply assume x lives in a subspace, we still do not know which one, and learning the subspace is a nonlinear problem. If the subspace is known, the recovery is indeed linear. Moving towards sparse models, everything becomes nonlinear. To learn a sparse (union of subspaces) model, one has to do something like dictionary learning (again nonlinear). But in this case the recovery is nonlinear as well (l1 minimization or something similar). More general models with unclear algebraic characterization such as low-dimensional manifolds require more powerful learning structures.
> > >
> > > Finally, let us look at the conditions you mention. Consider an observation operator A which returns a few entries of x. Its nullspace N(A) is spanned by the canonical basis vectors corresponding to the unobserved entries. If x belongs to a subspace S which intersects N(A) (for example, it contains one of the said canonical basis vectors), then x cannot be reconstructed from Ax. Suppose that this is only approximately the case: S contains one of those canonical basis vectors perturbed by a tiny amount of noise. Then in principle x can be reconstructed, and the reconstruction operator is F = U pinv(A U) so that FA is an oblique projection. If y = Ax and x \in S, clearly Fy = x. But pinv(AU) will have a very large singular value, so if y = Ax + noise, the noise will dominate the reconstruction. While this is different from what we do (our x is not at all in any subspace), it is another instance where oblique projections can be very bad, this time due to noise. To get an idea of how noise aggravates things, please take a look at the new results with added noise.
> > >
> > > >> “This fact is at the heart of low-rank matrix completion and it seems to contradict your claim about how difficult to can be to compute these projections. How do I interpret your claim in this setting?”
> > >
> > > Response: In the light of the above discussion, we respectfully disagree. Low-rank matrix completion relies on the fact that we can identify the right low-dimensional subspace spanned by rank-1 matrices given sufficient measurements. If this subspace is already known then we agree with the reviewer’s example, but this is not the gist of low-rank matrix completion: it would only allow reconstructing special low-rank matrices that are linear combinations of some fixed rank-1 matrices.
> > >
> > > Generally, in low-rank matrix completion, we do not know the low-rank matrix basis, which makes the problem nonlinear (analogously, we do not know the sparse support in sparse models). Identifying the basis of rank-1 matrices is analogous to support recovery with sparse priors. The algorithms for low-rank matrix recovery are therefore not linear: they use regularizers such as the nuclear norm optimized by nonlinear schemes such as iterative singular value thresholding.
> > >
> > > Further, because of the particular structure of the measurement operator A here (“return some entries”), we need conditions on x (the matrix to recover) related to the above example of subspaces with many zero entries. In particular, if the matrix is at once low-rank and sparse, it will be problematic for entrywise observations which will return those zeros with significant positive probability. That is why the guarantees in low-rank matrix completion from a few entries assume that the matrix is not simultaneously sparse and low rank (see, e.g., Section 1.1.1. and the paragraph before Theorem 1.3 in [1] where this is formulated as “incoherence of row and column spaces with the standard basis”). Clearly, even with many near-zero entries in the matrix the recovery is unstable.
> > >
> > > [1] Candès, E.J. and Recht, B., 2009. Exact matrix completion via convex optimization. Foundations of Computational mathematics, 9(6), p.717.

---

> > > ### Author Response · Authors · 2018-11-25
> > > **Nonlinearity is required (Part 2)**
> > >
> > > Let us now analyze your proposed reconstruction. We first look at the formula: \hat v_k = pinv(A U_k)y, which corresponds to the expansion coefficients of the oblique projection in Appendix A. In general, how well the oblique projection \hat z_k = U_k \hat v_k approximates P_{S_k} x depends on the smallest principal angle between the subspaces R(A^*) and S_k. In the interesting case where this angle is close to pi/2 (i.e., where we are getting information about x in R(A^*)^\perp = N(A)), the linear method fails spectacularly because the pseudoinverse explodes (please see further discussion below and numerical experiments).
> > >
> > > -- If S_k happens to lie completely within the nullspace of A, then the product A U_k is a zero matrix and pinv(A U_k) is also a zero matrix. Thus even if x in general has an arbitrarily large component in S_k, your estimate of this component will be zero.
> > > -- A more common case: If N(A) intersects S_k only trivially (only at origin), but the smallest principal angle between the two subspaces is small (i.e., the smallest singular value sigma_min of A U_k is small), then pinv(A U_k) will be very large (in any norm) since 1 / sigma_min is large, and the point (P_S^oblique) x will diverge to infinity. To see this geometrically, imagine that S_\lambda in Figure 8 is being rotated so that the angle between the R(A^*) and S_\lambda approaches pi/2. The oblique projection point will travel to infinity because the projection always take place along the line orthogonal to R(A^*) (along the nullspace of A).
> > >
> > > A naive proposal to fix this by choosing subspaces so that the R(A^*) and S_k are close is not useful because those subspaces give the same information as pinv(A). “Useful” subspaces reveal information about x in N(A) and those are precisely the ones that cause trouble. We want to choose the subspaces independently of A.
> > >
> > > Another proposal could be to regularize the pinv by strategies such as Tikhonov regularization, but these methods will not reinstate the nullspace components because x does not live in any subspace, and the overall reconstruction would again be forced to be in a certain subspace, as explained in more detail in what follows.
> > >
> > > Let us see how this shows up in your suggested minimization \min x \sum_k \|P_{S_k} x - \hat z_k \|_2^2 (with squared norm). Any solution to this convex problem satisfies (by setting the gradient to zero):
> > >
> > > \sum_k P_{S_k}^* (P_{S_k} \hat x - \hat z_k) = 0
> > >
> > > Using the fact that P_{S_k} is an orthogonal projection, hence self-adjoint (P_{S_k} = P_{S_k}^*) and idempotent (P_{S_k}^2 = P_{S_k}), and that \hat z_k already lives in S_k, we can write this as \sum_k (P_{S_k} \hat x - \hat z_k) = 0, or (dividing both sides by the total number of subspaces so that we can think in terms of averages):
> > >
> > > (1/K) ( \sum_k P_{S_k} ) \hat x  = (1/K) \sum_k \hat z_k
> > >
> > > For a large enough number of random subspaces K, the matrix R = (1/K) ( \sum_k P_{S_k} ) on the left-hand side becomes full rank. Since \hat z_k = U_k pinv(A U_k) A x (up to noise), the right-hand side can be written
> > >
> > > (1/K) \sum_k U_k pinv(A U_k) A x = {[ (1/K) \sum_k U_k pinv(A U_k) ] A} x.
> > >
> > > The row space of the matrix G =  [ (1/K) \sum_k U_k pinv(A U_k) ] A multiplying x on the rhs is the same as the row space of A, so G is low-rank (it is an oblique projection matrix on some subspace). This gives
> > >
> > > \hat x = inv(R) G x,
> > >
> > > which can only be a good estimate if x is already in the range of inv(R) G x (a subspace). But again, x is not constrained to any particular subspace.
> > >
> > > Any linear reconstruction, no matter how regularized, can only produce results in a fixed subspace with dimension at most the number of rows of A (for any matrix B, rank(BA) is at most the number of rows in A, so its column space is a fixed low-dimensional subspace). The nullspace of A dictates what can and what cannot be recovered. On the other hand, our method can easily provide information in the nullspace because it explores non-linear correlations between the nullspace and range space components of x (via a manifold model).
> > >
> > > To empirically support the mathematical fact that oblique projections and linear reconstruction can be arbitrarily bad, we simulate your proposed approach ( https://tinyurl.com/obliqueandprojnetfigure ). The code can be found at https://tinyurl.com/obliqueandprojnetcode . Note that the subspaces we use are the same that we used in the experiments in the manuscript.

---

> > > ### Author Response · Authors · 2018-11-25
> > > **Nonlinearity is required (Part 1)**
> > >
> > > >> “Thank you for the clarification on how the method works; this cleared up some things. However, it's still not clear to me why this should work. I agree with the other reviewer that the ensemble hypothesis is one potential explanation, but the paper would be strengthened by more depth in this regard.”
> > >
> > > Response: As we have explained in our response to the Reviewer 2 (part 2 of our response), while ensembling is a nice interpretation, the fact that a single network, a SubNet, performs just as well as many ProjNets shows that ensembling is not essential. The gist of our method is that projections are easier to estimate, but they require a nonlinear estimator. We elaborate this in detail in response to your comments below. Currently the 8 pages of the manuscript are tightly filled up with the problem description, mathematical motivations, intuitions, and numerical examples which show that the method beats strong baselines. We also added new explanations and figures in the appendix motivated by the reviewers’ comments. We feel that it would be challenging to add more material without removing some important parts and that further results on the various aspects of the method should be part of future publications.
> > >
> > > >> “It would also help to see some concreteness to some of the explanations. I find Appendix A and Figure 8 difficult to follow. Is \cal R(A*) the range of the adjoint of A? I can't find this defined anywhere. Likewise, I can't find a concrete definition of P_S^oblique.”
> > >
> > > Response: \cal R(A*) is indeed the range of the adjoint of A. Thank you for pointing out this omission. We have added the definition to Appendix A and Figure 8. The definition of P_S^oblique follows from the definition of an oblique projection with a given range and nullspace (and matches your \hat z_k below). We also added a few other clarifications in the Appendix which hopefully make it easier to follow.
> > >
> > > >> “Consider this comparison point. Let S_k be a random subspace and U_k be a basis spanning it. Then z_k := P_{S_k} x = U_k v_k for some coefficient vector v_k. Thus one estimator of z_k is simply \hat z_k = U_k \hat v_k where \hat v_k = pinv(AU_k)y. From these \hat z_k's I could then estimate x via
> > > \min x \sum_k \|P_{S_k} x - \hat z_k \|_F
> > >
> > > I think the above approach is consistent with the spirit of your method, but based on linear estimators of z_k instead of CNNs. But this raises several questions:
> > >
> > > 1. Is the \hat z_k above (which I think might correspond to your P_S^oblique) consistent with your appendix A claim that the oblique projection can be arbitrarily bad? I find this difficult to interpret.”
> > >
> > > Response: Indeed, your proposed \hat z_k is the oblique projection which is denoted P_{S}^{oblique} x in Appendix A and Figure 8. Further, your reconstruction (where we squared the norm):
> > >
> > > \min x \sum_k \|P_{S_k} x - \hat z_k \|_2^2
> > >
> > > is the same as our (2), without the regularizer and constraints. To see this equivalence, assume without loss of generality that the columns of U_k are orthonormal so that P_{S_k} = U_k U_k^T. Since \| . \|_2 is unitarily invariant, left multiplication by orthonormal U_k does not change it so we can write the minimization as \min_x \sum_k \|U_k^T x - \hat v_k \|_2^2. Noting that \hat v_k is our q_\lambda and U_k our B_\lambda, and stacking the terms in the sum, we get the data term in (2).
> > >
> > > And yes, \hat z_k can be arbitrarily bad. Let us try again to explain why this is the case (both mathematically and with numerical examples (see link https://tinyurl.com/obliqueandprojnetfigure ). In what follows N(A) will denote the nullspace of matrix A, R(A^*) the range of matrix A^*, where ^* denotes the adjoint (which is the transpose for real matrices). Superscript ^\perp denotes the orthogonal complement.
> > >
> > > First, we note that in underdetermined inverse problems y = Ax + n, the role of any regularizer is to provide information about x in the nullspace of A. The unknown vector x has a component along the nullspace of A and along its orthogonal complement, N(A)^\perp = R(A^*). The component along the orthogonal complement of N(A) is simply pinv(A)*y which is the orthogonal projection of x into R(A^*).
> > >
> > > The only situation where linear methods can provide this nullspace information is when x is constrained to a *known* subspace. In this case the reconstruction is given by the oblique projection U pinv(A U) y (where the columns of U span the subspace) and there is no need for random projections. But this is not useful for us, because our x does not live in any subspace, let alone a known one. It is well known that most interesting signal classes (natural images, biomedical images, seismic images, anything with singularities such as edges) are not efficiently modeled by subspaces. That is why modern methods rely on sparse models, low-rank models, manifold models, and other non-linear models.

---

> > > ### Author Response · Authors · 2018-11-25
> > > **Summary of our detailed responses below**
> > >
> > > TL;DR: Your proposed method is indeed equivalent to a linear oblique projection which we described in Appendix A. The oblique projection into a subspace can become arbitrarily bad when the subspace which we want to project into is not aligned well with the range of A^* (adjoint of A). In this response we explain why this is the case both mathematically and via numerical experiments for which we share the code.
> > >
> > > (A side remark: we hope that the reviewer was able to read our responses to all their other previous comments in Parts 2, 3 and 4 of our first response. Due to the way OpenReview displays comments, it may have been unclear that those were parts of our response. )

---

### Official Review · AnonReviewer1 · 2018-11-05
**novel method for inverse problems**

**Rating:** 7
**Confidence:** 4

**Review:**

This paper proposes a novel method of solving ill-posed inverse problems and specifically focuses on geophysical imaging and remote sensing where high-res samples are rare and expensive.
The motivation is that previous inversion methods are often not stable since the problem is highly under-determined.  To  alleviate these problems, this paper proposes a novel idea:
instead of fully reconstructing in the original space, the authors create reconstructions in projected spaces.
The projected spaces they use have very low dimensions so the corresponding Lipschitz constant is small.
The specific low-dimensional reconstructions they obtain are piecewise constant images on random Delaunay trinagulations. This is theoretically motivated by classical work (Omohundro'89) and has the further advantage that the low-res reconstructions are interpretable. One can visually see how closely they capture the large shapes of the unknown image.

These low-dimensional reconstructions are subsequently combined in the second stage of the proposed algorithm, to get a high-resolution reconstruction. The important aspect is that the piecewise linear reconstructions are now treated as measurments which however are local in the pixel-space and hence lead to more stable reconstructions.

The problem of reconstruction from these piecewise constant projections is of independent interest. Improving this second stage of their algorithm, the authors would get a better result overall. For example I would recommend using Deep Image prior as an alternative technique of reconstructing a high-res image from multiple piecewise constant images, but this can be future work.

Overall I like this paper. It contains a truly novel idea for an architecture in solving inverse problems. The two steps can be individually improved but the idea of separation is quite interesting and novel.

---

> ### Author Response · Authors · 2018-11-16
> **Thank you for your insights on extending our work**
>
> We are glad that the reviewer enjoyed the paper. Indeed one of the main ideas put forward is the separation into information that can be stably (but nonlinearly) extracted from the measurements in this very ill-posed, no ground truth regime, and information that requires a stronger regularizing idea which kicks in at stage 2. We find it encouraging that the reviewer’s comments on improving stage 2 are quite similar to our ideas on extending this work (we now mention this in the concluding remarks). Further, we now provide an additional discussion of why the method can work and why nonlinear regressors are necessary in Appendix A and an updated Section 3.1, as an effort to address the comments of other reviewers.

---

> > ### Comment · AnonReviewer1 · 2018-12-09
> > **post rebuttal**
> >
> > I read the reply to my review, the other reviews and the extended discussion on this paper. I am glad the OpenReview system is working well for this paper.
> >
> > My current vote is on accepting this paper given that there is clearly extensive work put into it and it contains several interesting novel ideas.

---

> > > ### Author Response · Authors · 2018-12-12
> > > **Thank you**
> > >
> > > Thank you for taking the time to read through all our responses. We are glad that you like our work.

---

### Official Review · AnonReviewer2 · 2018-11-07
**Interesting method, but limited demonstrations and unclear reason for working**

**Rating:** 6
**Confidence:** 4

**Review:**

Summary:
Given an inverse problem, we want to infer (x) s.t. Ax = y, but in situations where the number of observations are very sparse, and do not enable direct inversion. The paper tackles scenarios where 'x' is of the form of an image. The proposed approach is a learning based one which trains CNNs to infer x given y (actually an initial least square solution x_init is used instead of y).

The key insight is that instead of training to directly predict x, the paper proposes to predict different piecewise constant projections of x from x_init , with one CNN trained for each projection, each projection space defined from a random delaunay triangulation, with the hope that learning prediction for each projection is more sample efficient. The desired x is then optimized for given the predicted predicted projections.

Pros:
- The proposed approach is interesting and novel - I've not previously seen the idea of predicting different picewise constant projections instead of directly predicting the desired output (although using random projections has been explored)
- The presented results are quantitatively and qualitatively better compared to a direct prediction baseline
- The paper is generally well written, and interesting to read

Cons:
While the method is interesting, it is apriori unclear why this works, and why this has been only explored in context of linear inverse problems if it really does work.

- Regarding limited demonstration: The central idea presented here is is generally applicable to any per-pixel regression task. Given this, I am not sure why this paper only explores it in the particular case of linear inversion and not other general tasks (e.g. depth prediction from a single image). Is there some limitation which would prevent such applications? If yes, a discussion would help. If not, it would be convincing to see such applications.

- Regarding why it works: While learning a single projection maybe more sample efficient, learning all of them s.t. the obtained x is accurate may not be. Given this, I'm not entirely sure why the proposed approach is supposed to work. One hypothesis is that the different learned CNNs that each predict a piecewise projection are implicitly yielding an ensembling effect, and therefore a more fair baseline to compare would be a 'direct-ensemble' where many different (number = number of projections) direct CNNs (with different seeds etc.) are trained, and their predictions ensembled.


Overall, while the paper is interesting to read and shows some nice results in a particular domain, it is unclear why the proposed approach should work in general and whether it is simply implicitly similar to an ensemble of predictors.

---

> ### Author Response · Authors · 2018-11-16
> **Explanation for why it works and motivation for "linear" inverse problems (Part 1)**
>
> >> “Pros:
> - The proposed approach is interesting and novel - I've not previously seen the idea of predicting different picewise constant projections instead of directly predicting the desired output (although using random projections has been explored)
> - The presented results are quantitatively and qualitatively better compared to a direct prediction baseline
> - The paper is generally well written, and interesting to read”
>
> Response: We are glad that the reviewer found the paper interesting.
>
> >> “While the method is interesting, it is apriori unclear why this works, and why this has been only explored in context of linear inverse problems if it really does work."
>
> >> "Regarding limited demonstration: The central idea presented here is is generally applicable to any per-pixel regression task. Given this, I am not sure why this paper only explores it in the particular case of linear inversion and not other general tasks (e.g. depth prediction from a single image). Is there some limitation which would prevent such applications? If yes, a discussion would help. If not, it would be convincing to see such applications.”
>
> Response: While we agree with the reviewer that the central idea is more widely applicable, we wish to emphasize that what the reviewer calls a “particular case of linear inversion” covers a very large variety of practically relevant problems. The list includes super-resolution, deconvolution, computed tomography, inverse scattering, synthetic aperture radar, seismic tomography, radio-interferometric astronomy, and many other problems.
>
> Importantly, the fact that the forward problem is linear (which is why the corresponding inverse problems are unfortunately called linear) does not at all imply that the sought inverse map which we are trying to learn (the solution operator) is linear. The inverse map of interest will not be linear for anything but the simplest Tikhonov regularized solution (and variations thereof). For instance, if x is modeled as sparse in a dictionary, the inverse map is nonlinear even though the vast majority of inverse problems regularized by sparsity are “linear". The entire field of compressive sensing is concerned with linear inverse problems. With general manifold models for x, such as the one assumed in the paper, we depart further from linear inverse maps. We now state this more explicitly in Section 3.1 and a new Appendix A. The ability to adapt to such nonlinear prior models is part of the reason why CNNs perform well on related problems. Additionally, these nonlinear inverses may be arbitrarily ill-posed, which calls for ever more sophisticated regularizers. In this sense, we are looking at a very large class of hard, practically relevant problems, whose solution operators are nonlinear.
>
> While nothing prevents practical application of our proposed method to problems such as single-image depth estimation, one benefit of studying linear inverse problems is that as soon as we are in finite dimensions (e.g., a low-dimensional manifold in R^N and a finite number of measurements), and the forward operator is injective, Lipschitz stability is guaranteed (refer added citation: [1]). Injectivity can be generically achieved with a sufficient number of measurements that depends only on the manifold dimension.
>
> In applications such as depth estimation from a single image it is less straightforward to obtain similar guarantees. Namely, injectivity fails as one can easily construct cases where the same 2D depth map corresponds to multiple 2D images. So, while in practice our method might give good results, the justification would require additional work.

---

> > ### Author Response · Authors · 2018-11-16
> > **Explanation for why it works and motivation for "linear" inverse problems (Part 2)**
> >
> > >> “Regarding why it works: While learning a single projection maybe more sample efficient, learning all of them s.t. the obtained x is accurate may not be. Given this, I'm not entirely sure why the proposed approach is supposed to work. One hypothesis is that the different learned CNNs that each predict a piecewise projection are implicitly yielding an ensembling effect, and therefore a more fair baseline to compare would be a 'direct-ensemble' where many different (number = number of projections) direct CNNs (with different seeds etc.) are trained, and their predictions ensembled.”
> >
> > Response: Recall that we are in a regime where we do not have access to a large ground-truth training dataset and the measurements are very sparse. For this reason, we cannot hope to get a method that reconstructs all the details of x. This is the motivation to split the problem in two stages: in the first stage we only estimate “stable” information, by learning a collection of nonlinear, but stable maps from y (or pinv(A)*y, or its non-negative least squares reconstruction) to projections of x. As shown experimentally, this strategy outperforms the baseline which uses the exact same number of measurements and training samples. In fact, all ProjNets are trained using half the number of samples as the baseline (we now make this more explicit in the manuscript).
> >
> > In the second stage of computing x from the projections, in order to get a very accurate, detailed estimate, one would need to use more training samples, and those samples should correspond to ground truth images which we do not have. Furthermore, as Reviewer 1 suggests, this might involve new and better regularizers.
> >
> > We agree with the reviewer’s hypothesis that the different learned CNNs are implicitly yielding an ensembling effect—that is a nice interpretation of the proposed method. However, because the direct inverse map from y to x is highly unstable, we design a randomization mechanism which is better behaved than just training neural networks with different seeds. The instability of the full inverse map y -> x (or x0 -> x) will result in large systematic errors that will not average out. To illustrate this, per reviewer’s suggestion, we trained ten new direct networks and repeated the erasure experiments (Figures 5b, 12, 13) for the case when p=1/8. If, for example, we consider the image in Figure 5b, we find that 9/10 direct network reconstructions look almost the same as the poor reconstruction shown in the manuscript (see: https://tinyurl.com/direct-new-seeds ), while one reconstruction looks a bit closer to the true x, but still quite wrong (much more so than the reconstructions from the ProjNets). Our randomization scheme operates by providing random, low-dimensional targets that are stable and have low variance so that the resulting estimates are close to their true values and the subsequent ensembling mechanism is deterministic (in the sense that it does not rely on “noise”). We stress again that the total number of training samples used to train all ProjNets, or the single SubNet is the same or smaller than that used to train the direct baseline.
> >
> > Moreover, we point out that we train two different architectures—one that requires a different network for each subspace (ProjNet) and one that works for any subspace (SubNet). The success of SubNet and the fact that it outperforms the direct baseline suggests that the important idea is indeed that of estimating low-dimensional projections.
> >
> > Another important aspect of our choice of randomization is that it leads to interpretable, local measurements. These correspond to a new, equivalent forward operator B with favorable properties (see Section 3.2, 3.3 and Proposition 1). It would be hard to interpret the output of randomly initialized direct networks in a similar way (for example, it is not clear what we should expect the output distribution to be).
> >
> > [1] Stefanov, P. and Uhlmann, G., 2009. Linearizing non-linear inverse problems and an application to inverse backscattering. Journal of Functional Analysis, 256(9), pp.2842-2866.

---

> > > ### Comment · AnonReviewer2 · 2018-12-11
> > > **Updated rating following the paper revision.**
> > >
> > > I'd like to thank the authors for the revised version of the manuscript. I agree with the response that tackling linear inversion is of a more general interest than my initial review indicates, and is a good setting to study given the possibility of theoretical analysis. I also agree with the response the other review concern that the non-linearity is required for the inversion function, and also more positive about the presentation as the approach is presented much more clearly in the revised version.
> > >
> > > I am updating my rating primarily based upon the additional visualizations presented in the response regarding the performance of a simple ensemble method,  and qualitative results showing the proposed method does better empirically. However, I do not think these results and a corresponding discussion are currently in the revised manuscript, and the comparison to the simple ensemble method is purely qualitative - I strongly encourage the authors to incorporate these results/discussion in the final version, and also add quantitative comparison to average predictions obtained via an ensemble.

---

> > > > ### Author Response · Authors · 2018-12-12
> > > > **Thank you**
> > > >
> > > > Thank you for taking the time to read through our responses and for the positive assessment. We definitely intend to add the suggested information to the final version. We were perhaps a bit conservative trying to avoid a “significant” change.

---

### Author Response · Authors · 2018-11-16
**Summary of responses to reviewers**

We thank the reviewers for taking the time to read the paper and prepare their comments. All are informative and they made us aware of the parts of presentation that might have been confusing; we hope that our updates make the manuscript clearer.

With some of the comments, though, we have to respectfully disagree. We explain this in the responses to individual reviewers. Here we only summarize a few main points, before addressing the individual reviewers’ comments in detail.

-- In our method we solve a linear inverse problem y = Ax + n which is very ill posed, without having access to ground truth training data. To do so, we train a non-linear regressor (a neural net) which maps y to orthogonal projections of x into random subspaces with an arbitrarily chosen training dataset. To simplify network structure, we precompute x0 which can be an application of a pseudoinverse of A to y, a non-negative least squares solution or some other simple estimator. Importantly, because the measurements are few and the problem is very ill posed, x0 is a very bad estimate of x.

-- We do not project x0 into random subspaces as Reviewer 3 suggests—this is achieved by a simple linear operator and would be of limited interest. We rather compute *orthogonal* projections of x from x0. As we elaborate in the updated manuscript (see Appendix A) and in the response to Reviewer 3, this cannot be achieved by a linear operator and it requires training a nonlinear regressor (in our case, a neural network).

-- The term “linear inverse problems” only implies that the forward operators are linear. In most interesting applications, the inverse operators are arbitrarily nonlinear. This is the case already with standard sparsity-based methods. In our case, since we do not know where x lives, the nonlinear modeling is achieved by learning. Many, if not most practical imaging problems have (approximately) linear forward operators: examples are synthetic aperture radar, seismic tomography, radio-interferometric astronomy, MRI, CT, etc. While certainly many are only approximately linear (or fully nonlinear), linearization techniques are at the core of both practical algorithms and theoretical analysis. The latter is true even for questions of uniqueness and stability as discussed beautifully in [1]. In this sense we are looking at a very important and large class of nonlinear operators to be learned, and we do not see our discussion of linear inverse problems as a harsh limitation. That said, our method could be applied to other problems such as depth sensing, as suggested by Reviewer 2, but the justification would require additional work. For example, the Lipschitz stability (which we have per [1]) would not be guaranteed. The fact that an inverse exists for the imaging tasks we consider is given by injectivity on  \mathcal{X}, which is a low-dimensional structure (a manifold) embedded in R^N. In the original manuscript this assumption was in a footnote which is now expanded into a short discussion in Section 3.1. We elaborate this further in the response to Reviewer 2.

-- Our method can be interpreted as a randomization or an ensembling method. But unlike strategies such as randomizing the seed when training many neural networks to directly estimate x, which will be hampered by the instability of the problem and the fact that we do not have ground truth data, we use a particular randomization scheme where we randomize the learning target. That way we a) have a clear model for randomization which tells us exactly how to use the individual projection estimates, and b) make each individual member of the problem ensemble stable.

[1] Stefanov, P. and Uhlmann, G., 2009. Linearizing non-linear inverse problems and an application to inverse backscattering. Journal of Functional Analysis, 256(9), pp.2842-2866.

---

### Meta-Review · Area_Chair1 · 2018-12-14
**interesting direction for inverse problems**

**Confidence:** 3
**Recommendation:** Accept (Poster)

**Metareview:**

This paper proposes a novel method of solving inverse problems that avoids direct inversion by  first reconstructing various piecewise-constant projections of the unknown image (using a different CNN to learn each)  and then combining them via optimization to solve the final inversion.
Two of the reviewers requested more intuitions into why this two stage process would  fight the inherent ambiguity.
At the end of the discussion, two of the three reviewers are convinced by the derivations and empirical justification of the paper.
The authors also have significantly improved the clarity of the manuscript throughout the discussion period.
It would be interesting to see if there are any connections between such inversion via optimization with deep component analysis methods, e.g. “Deep Component Analysis via Alternating Direction Neural Networks
” of Murdock et al. , that train neural architectures to effectively carry out the second step of optimization, as opposed to learning  a feedforward mapping.